

# Equivalent Hazard Magnitude Scale

Yi Victor Wang[1], Antonia Sebastian[1]

[1] Department of Geological Sciences, University of North Carolina at Chapel Hill, Chapel Hill, NC, 27514, United States of America

*Correspondence to*: Yi Victor Wang (y.v.wang@unc.edu)

**Abstract.** Hazard magnitude scales are widely adopted to facilitate communication regarding hazard events and the corresponding decision making for emergency management. A hazard magnitude scale measures the strength of a hazard event considering the natural forcing phenomena and the severity of the event with respect to average entities at risk. However, existing hazard magnitude scales cannot be easily adapted for comparative analysis across different hazard types. Here, we

propose an equivalent hazard magnitude scale, called the Gardoni Scale after Professor Paolo Gardoni, to measure hazard strength across multiple types of hazards. Using global historical records of hazard magnitude indicators and impacts of events of 12 hazard types from 1900 to 2020, we demonstrate that an equivalent hazard magnitude on the Gardoni Scale can be derived as correlated with the expectation of an impact metric of hazard event. In this study, we model the impact metric as a function of fatalities, total affected population, and total economic damage. Our results show that hazard magnitudes of events

can be evaluated and compared across hazard types. For example, we find that tsunami and drought events tend to have large hazard magnitudes, while tornadoes are relatively small in terms of hazard magnitude. In addition, we demonstrate that the scale can be used to evaluate hazard equivalency of historical events. For example, we show that the hazard magnitude of the February 2021 North American cold wave event affecting the southern states of the United States of America was equivalent to the hazard magnitude of Hurricane Harvey in 2017 or a magnitude 7.5 earthquake. Future work will expand the current

study in hazard equivalency to modelling of local intensities of hazard events and hazard conditions within a multi-hazard context.

## 1 Introduction

Natural hazards pose significant challenges to human societies around the world. Between 2000 and 2020, natural hazard events caused over 130 billion dollars in losses, 64 695 fatalities, and affected more than 196 million people on average each

year (Guha-Sapir et al., 2021). Hazard events with a strong natural force, such as earthquakes, floods, and forest fires, can inflict heavy losses to communities when vulnerable living beings and properties of value are exposed to the natural forces of these events. To measure the size of a hazard event in terms of its impacts, several research teams have proposed impact scales, including the Bradford disaster scale (Keller et al., 1992; 1997), unified localizable crisis scale (Rohn and Blackmore, 2009; 2015), disaster impact index (Gardoni and Murphy, 2010), and cascading disaster magnitude (Alexander, 2018). These impact

scales take into consideration the effect of hazard strength. However, there is a qualitative difference between a hazard impact





scale and a hazard strength scale: while a hazard strength scale such as the earthquake moment magnitude can be used to indicate the degree of natural force of an event, the impacts of event are also determined by the exposure and vulnerability of entities such as individuals, communities, and infrastructures to the event. Thus, while a hazard event may have a large value on a hazard strength scale, it may also be associated with a small or zero measure on an impact scale due to low exposed value

and/or low vulnerability. Simultaneously, a hazard with events frequently occurring in a geographic region with a small value on a hazard strength scale, may be ranked high in terms of its impact if there have been a large number of exposed entities and/or high vulnerability to that type of hazard historically. This makes it difficult to use impact scales to compare the hazard strength across natural hazard types.

Because most of the hazard-related concepts such as hazard exposure, hazard vulnerability, and hazard resilience are only
meaningful when considered with respect to a spectrum of hazard strength, studies on hazard strength indicators and scales provide the foundational knowledge and frameworks in the field of natural hazards. In addition, hazard scientists have long called for separation of natural forcing phenomena (Bensi et al., 2020) from the study of disasters to better understand the causes of impacts rooted in the social and economic fabrics of entities exposed to natural hazards (e.g., O'Keefe et al. 1976; Wisner et al. 2004). In this regard, quantifying hazard strength helps separate the natural force from other environmental,
societal, and infrastructural impacts to facilitate scientific understanding of natural hazard phenomena for disaster risk reduction, especially within a multi-hazard context. Yet, despite the large volume of research that focuses on hazard strength for singular natural hazard types such as earthquake (e.g., Wood and Neumann, 1931; Richter, 1935; Kanamori, 1977; Katsumata, 1996; Grünthal, 1998; Wald et al., 2006; Rautian et al., 2007; Serva et al., 2016), tropical cyclone (e.g., Simpson and Saffir, 1974; Bell et al., 2000; Emanuel, 2005; Powell and Reinhold, 2007; Hebert et al., 2008), tornado (e.g., Fujita, 1971;
1981; Meaden et al., 2007; Potter, 2007; Dotzek, 2009), and drought (e.g., Palmer, 1965; 1968; Shafer and Dezman, 1982; McKee et al., 1993; Byun and Wilhite, 1999; Shukla and Wood, 2008; Hunt et al., 2009), few have quantified or modelled hazard strength across multiple hazard types for cross-hazard comparison.

To enable evaluation of event-wise hazard strengths across different hazard types, in this article, we propose a multi-hazard *equivalent hazard magnitude scale – the Gardoni Scale –* for natural hazards. The proposed scale is named after the Alfredo
H. Ang Family Professor Paolo Gardoni (2017; 2019) at the University of Illinois at Urbana–Champaign (Gardoni and Murphy, 2013; 2014; 2020). Because hazard strength is correlated with hazard impacts given average exposed value and vulnerability of considered entities, the expectation of a metric of observed impacts of hazard events can be used to calibrate models for deriving equivalent hazard magnitudes on the Gardoni Scale (Hillier et al. 2015; Hillier and Dixon 2020; Wang and Sebastian, 2021b). In this article, a quantitative modelling methodology based on a principal component analysis (PCA) and a set of linear
regressions is developed to construct the impact metric and derive equivalent hazard magnitudes on the Gardoni Scale. The impact metric is a function of three impact variables, i.e., fatality, total affected population, and total damage in 2019 USD. We use historical data from the EM-DAT International Disaster Database (Guha-Sapir et al., 2021) on hazard magnitude indicators and impact variables of global hazard events of 12 natural hazard types from 1900 to 2020 to calibrate the





quantitative models. To demonstrate the value of the proposed scale, we apply it to discuss the equivalent magnitudes of recent

hazard events in the southern United States of America (USA).

In subsequent sections, we first review issues associated with the existing scales for natural hazard events to provide the theoretical background of this study. We then introduce our methodology, including data processing, to derive equivalent hazard magnitude on the Gardoni Scale. Next, we lay out the results of applying our methodology and compare natural hazard types regarding derived equivalent hazard magnitudes. Finally, we discuss the potential contributions and limitations of the

presented study before concluding the article.

## 2 A Problem of Scales

In natural hazards research, theoretical frameworks are based on basic concepts, such as hazard, impact, exposure, vulnerability, and resilience, that often have overlapping or discipline-specific definitions (see, e.g., Klijn et al., 2015). These inconsistencies across disciplines may result in confusion in quantitative modelling. To reduce conceptual confusion, we first

outline the theoretical framework and clarify the meanings of basic concepts used in this study. Herein, the impacts of a natural hazard event, such as an earthquake, tropical cyclone, or tornado, are the result of natural strength of hazard agent, value of entities exposed to the event, and susceptibility of the exposed entities to hazard impacts (Nigg and Mileti, 1997; Coburn and Spence, 2002; Wisner et al., 2004; Dilley et al. 2005; McEntire, 2005; Peduzzi et al. 2005; Adger, 2006; Burton, 2010; Lindell, 2013; Birkmann et al., 2014; Highfield et al., 2014; van de Lindt et al., 2020). As shown in Fig. 1, hazard strength of an event

is one of the main drivers, albeit not the sole driver, of impacts.

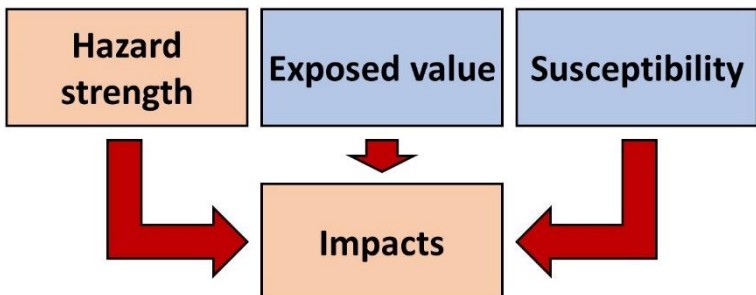

**Figure 1: Hazard event impacts as the result of hazard strength, exposed value, and susceptibility of exposed entities.**

Hazard strength is often referred to as the hazard magnitude or hazard intensity (Blong, 2003; Alexander, 2018). However, these two things are not equivalent. A hazard magnitude is a measure of the size of, or the total energy involved in, the entirety

of a hazard event (Blong, 2003; Alexander, 2018). Examples of hazard magnitude scale include earthquake Richter magnitude (Richter, 1935), earthquake moment magnitude (Kanamori, 1977), Abe tsunami magnitude (Abe, 1979), Murty–Loomis tsunami magnitude (Murty and Loomis, 1980), landslide magnitude (Arbanas and Arbanas, 2015), earthquake-triggered landslide magnitude (Tanyaş et al., 2018), volcanic explosivity index (Newhall and Self, 1982), Pyle volcanic eruption



magnitude (Pyle, 1995), accumulated cyclone energy index (Bell et al., 1999), tropical cyclone power dissipation index
(Emanuel, 2005), and solar radiation storm scale [Space Weather Prediction Center (SWPC), 2011].

In contrast, hazard intensity often refers to the hazard strength of an event with respect to a given location or area and/or a moment or period. Recently, Wang and Sebastian (2021b) identified two definitive dimensions, i.e., the spatial and temporal dimensions, to categorize existing hazard strength indicators and scales. With respect to the spatial dimension, a hazard strength scale is categorized as *agential* if it indicates the size of an event within its entire spatial range and *locational* if it is given for
a set of locations within the spatial range of an event. Likewise, a *durational* hazard strength scale corresponds to the entire duration of an event, while a *momental* hazard strength scale corresponds to a set of moments within the duration of an event. Considering both the spatial and temporal dimensions, we can therefore categorize hazard strength scales into four types, i.e., the *agential-durational scale*, the *locational-durational scale*, the *agential-momental scale*, and the *locational-momental scale*. In hazard literature, hazard magnitude scales can typically be categorized as agential-durational scales, while hazard intensity
scales correspond to locational-durational, agential-momental, or locational-momental scales. For example, the previously mentioned hazard magnitude scales are all agential-durational, whereas the earthquake modified Mercalli intensity scale (Wood and Neumann, 1931; Wald et al., 2006), integrated tsunami intensity scale (Lekkas et al., 2013), and drought magnitude (McKee et al., 1993) are locational-durational; the volcanic Fedotov intensity scale (Fedotov, 1985), Saffir–Simpson hurricane wind scale [National Hurricane Center (NHC) and Central Pacific Hurricane Center (CPHC), 2021], and geomagnetic storm
scale [Space Weather Prediction Center (SWPC), 2011] are agential-momental; and the tornado Fujita scale (Fujita, 1971), hailstorm intensity scale [The Tornado and Storm Research Organisation (TORRO), 2021], and Palmer drought severity index (Palmer, 1965) are locational-momental (Wang and Sebastian, 2021b). Given that an agential-durational hazard strength involves the entire spatial and temporal ranges of an event, with sufficient data, a locational-durational, agential-momental, or locational-momental hazard strength measure can be aggregated to form an agential-durational hazard strength measure of the
event. In this study, we use term "hazard magnitude" to refer to an agential-durational hazard strength of an event.

In addition to the hazard strength of an event, the value of exposed entities and susceptibility of the exposed entities also contribute to the impacts of the event. In hazard literature, the term "exposure" can be used to refer to either the exposed value or a quantity as the result of integration of hazard strength and exposed value (see, e.g., Dilley et al., 2005; Peduzzi et al., 2009; Klijn et al., 2015; Wang et al., 2020a; Tate et al., 2021; Wang and Sebastian, 2021a). There are also hazard exposure scales,
such as the Northeast snowfall impact scale (Kocin and Uccellini, 2004) and regional snowfall index (Squires et al., 2014) for snowstorms, specifically developed to quantify the distribution of hazard strength and exposed value associated with an event. Beside exposed value in Fig. 1, susceptibility of exposed entities in hazard literature may also be referred to or modelled in terms of fragility or vulnerability (e.g., Gardoni et al., 2002; 2003; Choe et al., 2007; Zhong et al., 2008; Huang et al., 2010; Wang et al., 2020b). The inverse of susceptibility is usually conceptualized as part of the domain of resilience (Holling, 1973;
Bruneau et al., 2003; Cutter et al., 2010; Alexander, 2013b; Ayyub, 2014; Dahlberg et al., 2015; Edwards, 2015; Yodo and Wang, 2016; Sharma et al., 2018; Logan and Guikema, 2020). In this study, we only focus on quantification of agential-durational hazard strengths of events on the proposed equivalent hazard magnitude Gardoni Scale across multiple hazard types.





## 3 Methodology

### 3.1 Data

To quantify hazard strength in terms of equivalent hazard magnitude, we used historical data on hazard impacts of hazard events to calibrate models. Because different protocols for data collection regarding different types of natural hazards may result in bias in model calibration, we only used data gathered from one database, i.e., the EM-DAT database (Guha-Sapir et al. 2021). To be included in the EM-DAT database, a hazard event needed to meet at least one of three criteria, i.e., 10 or more human fatalities, 100 or more people affected by the event, or a declaration of a state of emergency or an appeal for international

assistance by a country (Guha-Sapir et al. 2021). For this study, we downloaded the entire EM-DAT datasets on all types of natural hazards. However, due to a lack of records of hazard magnitude indicators of events for some hazard types, we only included 12 hazard types and kept one magnitude indicator for each hazard type. In addition, we removed all data points with missing values of hazard magnitude indicators from our final datasets.

The 12 considered hazard types, with their corresponding hazard magnitude indicators listed in parentheses, include: 1) cold

wave (minimum temperature in °C); 2) convective storm (peak gust wind speed in km h$^{-1}$); 3) drought (total affected area in km$^2$); 4) earthquake (Richter magnitude); 5) extra-tropical storm (peak gust wind speed in km h$^{-1}$); 6) flash flood (total flooded area in km$^2$); 7) forest fire (total burnt area in km$^2$); 8) heat wave (maximum temperature in °C); 9) riverine flood (total flooded area in km$^2$); 10) tornado (peak gust wind speed in km h$^{-1}$); 11) tropical cyclone (maximum sustained wind speed in km h$^{-1}$); and 12) tsunami (earthquake Richter magnitude). We removed data points with questionable values of hazard magnitude

indicators from our datasets. For cold wave events, we only included data points with a minimum temperature ≤0 °C; for convective storms, we only considered data points with a peak gust wind speed ≥60 km h$^{-1}$; for forest fires, we only included data points with a burnt area ≤200 thousand km$^2$; for heat wave events, we only considered data points with a maximum temperature ≥35 °C and ≤57 °C; for tornadoes, we only included data points with a peak gust wind speed ≥100 km h$^{-1}$; and for tsunami, we only considered data points with an earthquake Richter magnitude ≥6.

For regression modelling, we further logarithmically transformed values of hazard magnitude indicators to fit within the range $(-\infty, \infty)$ when necessary. Hazard magnitude indicators that were not logarithmically transformed included minimum temperature of cold waves, Richter magnitude of earthquakes, maximum temperature of heat waves, and earthquake Richter magnitude of tsunami. Cold wave and heat wave events were excluded from logarithmic transformations because Celsius temperature has a range $[-273.15, \infty)$ with its lower bound, $-273.15$, far away from 0. In the meantime, the range of an

earthquake Richter magnitude is already a desired $(-\infty, \infty)$.

### 3.2 Impact Metric

In this study, we designed the impact metric as the principal component (Jolliffe, 2002; Jolliffe and Cadima, 2016) of three logarithmically transformed and standardized impact variables. The selected three impact variables represent three major impact dimensions as defined by the EM-DAT database (Guha-Sapir et al. 2021). The first impact variable, fatality, indicates



the number of people killed in a hazard event. The second impact variable, total affected population, refers to the sum of numbers of residents injured, made homeless, or affected but not killed by the hazard event. The third impact variable, total damage, indicates the total amount of damage to property, crops, and livestock in 2019 USD caused by the hazard event. Values of impact variables were first logarithmically transformed to be within the range $(-\infty, \infty)$. The means and standard deviations of the logarithmically transformed impact variables were then applied to standardize the logarithmically transformed

impact variables (see Table 1) with the formula

$$IV = \frac{\ln(IVO) - \mu_{\ln IV}}{\sigma_{\ln IV}},$$      (1)

where $IV$ denotes the logarithmically transformed and standardized impact variable, $IVO$ is the original impact variable, $\mu_{\ln IV}$ and $\sigma_{\ln IV}$ are respectively the mean and standard deviation of the logarithmically transformed impact variable. The principal component (Jolliffe, 2002; Jolliffe and Cadima, 2016) of the three logarithmically transformed and standardized impact

variables corresponds to the dimension, along which the variation of data points is preserved to the largest extent in the three-dimensional vector space. The principal component also shows the direction of the eigenvector associated with the largest eigenvalue with respect to the covariance matrix of the three transformed impact variables. Each data point represents the impact of one hazard event experienced by one country (see supplementary material Video S1).

**Table 1: Means and standard deviations of original and logarithmically transformed impact variables used in the study[a].**

| Variable | Unit | Original mean | Original standard deviation | Logarithmically transformed mean | Logarithmically transformed standard deviation |
|---|---|---|---|---|---|
| Fatality | People | $1.31\times10^3$ | $1.18\times10^4$ | 3.3892 | 2.1999 |
| Total affected population | People | $1.38\times10^6$ | $9.47\times10^6$ | 10.4116 | 3.1618 |
| Total damage | 1 thousand 2019 USD | $1.36\times10^6$ | $8.45\times10^6$ | 11.1889 | 2.6304 |

[a]This table corresponds to supplementary material Data S1.



To reduce the bias caused by factors of exposed value and susceptibility of exposed entities (Fig. 1), we included all available data points at the country–year level for countries around the world and hazard events from a long period of 1900–2020 to

construct impact metric for the 12 considered hazard types. For derivation of impact metric, we only kept data points ($n = 1\,470$) without any missing values of the impact variables. A PCA was then conducted to determine the weights of transformed and standardized impact variables within the impact metric. These weights were the eigen values associated with the principal component of the transformed and standardized impact variables (Jolliffe, 2002; Jolliffe and Cadima, 2016). The resulting formula for impact metric is

$IM = 0.6158 IV_{\mathrm{F}} + 0.6215 IV_{\mathrm{TA}} + 0.4843 IV_{\mathrm{TD}}$ ,                                                    (2)

where $IM$ denotes the impact metric and $IV_{\mathrm{F}}$, $IV_{\mathrm{TA}}$, and $IV_{\mathrm{TD}}$ refer respectively to the transformed and standardized impact variables of fatality, total affected population, and total damage.

### 3.3 Missing Values and Data Aggregation

With the same data points for derivation of impact metric, we also calibrated six simple linear regression models and three bi-

variate linear regression models. These regression models were created to fill in missing values of impact variables for data points with at most two empty entries among the three impact variables. Within each of these nine linear regression models, the dependent variable is one of the three impact variables. For each of the six simple linear regression models, the independent variable is one of the two impact variables that are not used as the dependent variable. The simple linear regression models have the form

$IV_1 = a_1 + b_1 IV_2 + \sigma_1 \varepsilon$ ,                                                    (3)

where $a_1 = 0$ and $b_1$ are two model coefficients, $IV_1$ and $IV_2$ are two considered transformed and standardized impact variables, $\sigma_1$ is the dispersion parameter, and $\varepsilon$ is a standard normal random variable. The statistics of parameters of these simple linear regression models are shown in Table 2. Per the three bivariate linear regression models, the independent variables are the two impact variables other than the one used as the dependent variable. The formula for the bivariate linear

regression models is

$IV_1 = a_2 + b_2 IV_2 + c_2 IV_3 + \sigma_2 \varepsilon$ ,                                                    (4)

where $a_2 = 0$, $b_2$, and $c_2$ are three model coefficients, $IV_3$ is the third transformed and standardized impact variable, and $\sigma_2$ is the dispersion parameter. Table 3 lists the statistics of parameters of the bivariate linear regression models. By applying the derived linear regression models, we filled in the missing values of data points. We then aggregated the country–year data

points event-wise and reached a total of $3\,844$ data points, each representing one unique hazard event.



**Table 2: Statistics of parameters of six simple linear regression models for filling in missing values of impact variables[a].**

| Model number | Dependent variable | Independent variable | $b_1$ | $\sigma_1$ |
|---|---|---|---|---|
| I1 | Fatality | Total affected population | 0.5096 (0.0224) | 0.8604 (0.0159) |
| I2 | Fatality | Total damage | 0.2802 (0.0250) | 0.9599 (0.0177) |
| I3[b] | Total affected population | Fatality | 0.5096 (0.0224) | 0.8604 (0.0159) |
| I4 | Total affected population | Total damage | 0.2948 (0.0249) | 0.9556 (0.0176) |
| I5[c] | Total damage | Fatality | 0.2802 (0.0250) | 0.9599 (0.0177) |
| I6[d] | Total damage | Total affected population | 0.2948 (0.0249) | 0.9556 (0.0176) |

[a]This table corresponds to supplementary material Data S2; $R^2$s are included in Fig. 2; standard errors are in the parentheses; estimations of $b_1$ and $\sigma_1$ are all significant at $p < 10^{-20}$.

[b]Models I1 and I3 share the same model parameters and $R^2$.

[c]Models I2 and I5 share the same model parameters and $R^2$.

[d]Models I4 and I6 share the same model parameters and $R^2$.

**Table 3: Statistics of parameters of three bivariate linear regression models for filling in missing values of impact variables[a].**

| Model number | Dependent variable | Independent variable 1 | Independent variable 2 | $b_2$ | $c_2$ | $\sigma_2$ |
|---|---|---|---|---|---|---|
| I7 | Fatality | Total affected population | Total damage | 0.4676 (0.0232) | 0.1423 (0.0232) | 0.8496 (0.0157) |
| I8 | Total affected population | Fatality | Total damage | 0.4633 (0.0230) | 0.1650 (0.0230) | 0.8457 (0.0156) |
| I9 | Total damage | Fatality | Total affected population | 0.1755 (0.0286) | 0.2054 (0.0286) | 0.9435 (0.0174) |

[a]This table corresponds to supplementary material Data S3; $R^2$s are included in Fig. 2; standard errors are in the parentheses; estimations of

$b_2$, $c_2$, and $\sigma_2$ are all significant at $p < 10^{-8}$.


## 3.4 Regression Models for Individual Hazards

For each of the 12 considered hazard types, we calibrated one simple linear regression model to establish the relationship between hazard magnitude indicator and hazard impact metric. In general, such a regression model can be written as

$$IM = a_3 + b_3 MI + \sigma_3 \varepsilon ,\tag{5}$$

where $a_3$ and $b_3$ are two model coefficients, $MI$ denotes hazard magnitude indicator, and $\sigma_3$ is the dispersion parameter. The statistics of parameters of these 12 regression models are listed in Table 4. Parameters of all linear regression models involved in this study were determined with a maximum likelihood approach based on Raphson's algorithm (Raphson, 1697; Wang et al., 2019; Wang 2020). For each regression model, the standard errors of parameter estimates were derived from the main diagonal of the covariance matrix of model parameters computed as the negative inverse of the observed Fisher information

matrix.



**Table 4: Statistics of parameters of 12 simple linear regression models for deriving equivalent hazard magnitudes[a].**

| Model number | Hazard | $a_3$ | $b_3$ | $\sigma_3$ |
|:---:|:---:|:---:|:---:|:---:|
| M1 | Cold wave | −0.2404 (0.2171) | −0.0111 (0.0080) | 0.8595 (0.0726)*** |
| M2 | Convective storm | −7.5637 (2.1192)* | 1.3755 (0.4309)* | 0.7812 (0.0977)*** |
| M3 | Drought | −0.8833 (0.4691) | 0.2206 (0.0524)** | 1.0162 (0.1083)*** |
| M4 | Earthquake | −3.3328 (0.2308)*** | 0.4484 (0.0361)*** | 1.2464 (0.0246)*** |
| M5 | Extra-tropical storm | −12.2505 (6.6008) | 2.2827 (1.2965) | 1.3672 (0.1973)*** |
| M6 | Flash flood | −1.0275 (0.2244)*** | 0.0701 (0.0238)* | 0.9417 (0.0392)*** |
| M7 | Forest fire | −1.6116 (0.2221)*** | 0.1131 (0.0355)* | 0.8147 (0.0568)*** |
| M8 | Heat wave | −0.9524 (1.3678) | 0.0243 (0.0310) | 1.3297 (0.1002)*** |
| M9 | Riverine flood | −1.5284 (0.1349)*** | 0.1226 (0.0133)*** | 1.0140 (0.0209)*** |
| M10 | Tornado | −1.7272 (1.5488) | 0.1683 (0.2920) | 0.8511 (0.0784)*** |
| M11 | Tropical cyclone | −4.2569 (0.6510)*** | 0.8016 (0.1273)*** | 1.1719 (0.0326)*** |
| M12 | Tsunami | −7.0781 (2.0108)* | 0.9681 (0.2528)** | 1.2054 (0.1484)*** |

[a]This table corresponds to supplementary material Data S4; $R^2$s are included in Fig. 3; standard errors are in the parentheses.

$*p < 10^{-2}$; $**p < 10^{-3}$; $***p < 10^{-5}$.





### 3.5 Equivalent Hazard Magnitude Formula

To present equivalent hazard magnitude roughly within the range of [0, 10], we applied a linear transformation to the point estimate of impact metric

$$EM = \widehat{E}(IM) \times 2 + 5 \ , \tag{6}$$

where *EM* refers to the equivalent hazard magnitude and $\widehat{E}(\cdot)$ denotes the point estimate of expectation. The derived equivalent hazard magnitudes for all data points are recorded in supplementary material Data S6.

**4 Results**

### 4.1 Model Calibration

Visualization of the distribution of data points with respect to impact variables and impact metric (Figs. 2a, 2d, 2h, and 2m) shows that the empirical marginal distributions of the logarithmically transformed and standardized impact variables and the impact metric are approximately Gaussian. The standardized natural logarithms of impact variables are positively correlated

with each other (Figs. 2c, 2f, and 2g). Results of the bivariate linear regression modelling indicate that each of the standardized natural logarithms of impact variables is positively associated with the other two logarithmically transformed and standardized impact variables with a relatively medium $R^2$ (Figs. 2b, 2e, and 2i). These results provide justifications for leveraging data on some impact variables to interpolate missing values of other impact variables. Meanwhile, Figs. 2j–2l show that there are positive correlations between the impact metric and each of the standardized natural logarithms of impact variables with a

relatively large $R^2$. This result suggests the appropriateness of using as the impact metric the principal component of the three logarithmically transformed and standardized impact variables.





**Figure 2: Impact variables and impact metric.** (a) Histogram of impact variable fatality. (b) Fatality regressed on total affected population and total damage in 2019 USD with a multiple linear regression. (c) Total affected population regressed on fatality with a simple linear regression. (d) Histogram of impact variable total affected population. (e) Total affected population regressed on fatality and total damage in 2019 USD with a multiple linear regression. (f) Total damage in 2019 USD regressed on fatality with a simple linear regression. (g) Total damage in 2019 USD regressed on total affected population with a simple linear regression. (h) Histogram of impact variable total damage in 2019 USD. (i) Total damage in 2019 USD regressed on fatality and total affected population with a multiple linear regression. (j) Impact metric regressed on fatality with a simple linear regression. (k) Impact metric regressed on total affected population with a simple linear regression. (l) Impact metric regressed on total damage in 2019 USD with a simple linear regression. (m) Histogram of impact metric.





Results of calibration of linear regression models for 12 individual hazards (Fig. 3 and Table 4) show that the direction of coefficient of hazard magnitude indicator in each model is consistent with expectation. Unlike usual regression models that pursue precision, it is also expected that, in this study, the results of the regression models for individual hazards will show a wide spread of data points with respect to hazard magnitude indicator with a relatively small $R^2$ as in Fig. 3. This is because

the objective of this study is not to model or predict hazard impacts of an event, but to quantify the agential-durational hazard strength of the event. As suggested in Fig. 1, exposed value and susceptibility of exposed entities are two other main drivers of hazard impacts. The variations of data points with respect to hazard magnitude indicators in Fig. 3 indicate the significance of studying exposed value and susceptibility for disaster risk reduction. In the meantime, however, Fig. 3 also shows that the proposed methodology for deriving an equivalent hazard magnitude of an event is functional and effective in separating the

natural force, manifested in hazard strength, from other factors of impacts of natural hazard events to support studies on exposed value and susceptibility. In particular, the estimates of coefficients of hazard magnitude indicators for convective storm (Fig. 3b), drought (Fig. 3c), earthquake (Fig. 3d), flash flood (Fig. 3f), forest fire (Fig. 3g), riverine flood (Fig. 3i), tropical cyclone (Fig. 3k), and tsunami (Fig. 3l) are all statistically significant at $p < 10^{-2}$ (Table 4). This suggests that the derived equivalent hazard magnitudes on the Gardoni Scale for these hazard types are highly reliable. With the proposed

methodology, as a result, we can plot all data points onto one figure (Fig. 4) to compare equivalent hazard magnitudes of events across different hazard types on the Gardoni Scale.



**Figure 3: Simple linear regressions on impact metric against magnitude indicators.** Impact metric is regressed on **(a)** minimum temperature of cold wave; **(b)** peak gust wind speed of convective storm; **(c)** total affected area of drought; **(d)** Richter magnitude of earthquake; **(e)** peak gust wind speed of extra-tropical storm; **(f)** total flooded area of flash flood; **(g)** total burnt area of forest fire; **(h)** maximum temperature of heat wave; **(i)** total flooded area of riverine flood; **(j)** peak gust wind speed of tornado; **(k)** maximum sustained wind speed of tropical cyclone; **(l)** earthquake Richter magnitude of tsunami. Solid lines are regression lines. Shaded areas are the 95% confidence intervals of the corresponding regression lines.




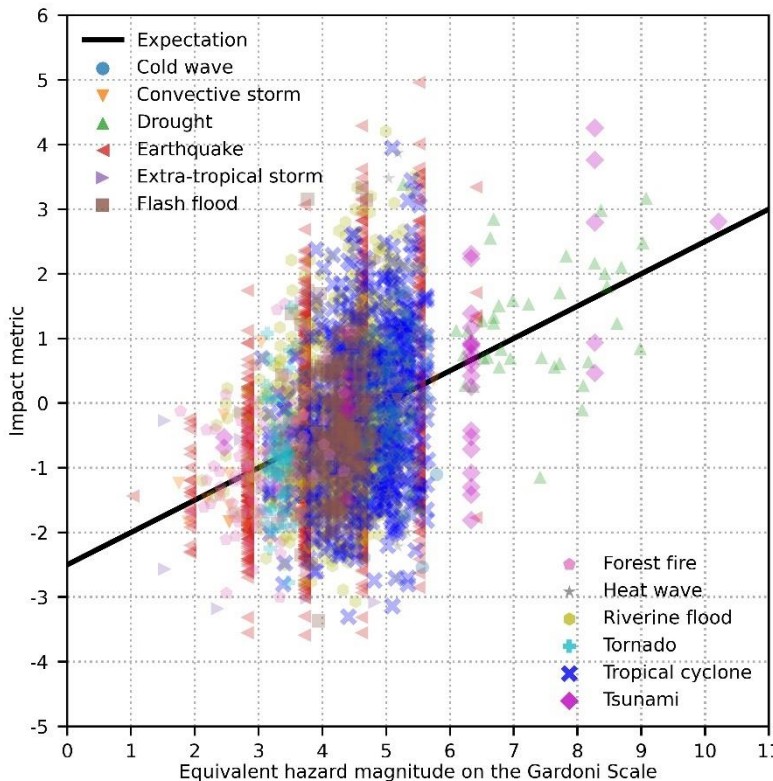

**Figure 4: Impact metric versus equivalent hazard magnitude on the Gardoni Scale.** The expectation line shows values of the expected impact metric with respect to equivalent hazard magnitude.



### 4.2 Comparisons of Hazard Magnitudes

As each data point on Fig. 4 corresponds to a record of hazard event in the EM-DAT database, all plotted data points are associated with impacts beyond a certain threshold defined by the database (Guha-Sapir et al. 2021). Therefore, a large quantity

of hazard events with small or zero impacts are excluded. Although a few of these excluded hazard events may have a large hazard magnitude, it is rational to assume that most of the excluded data points are with a small hazard magnitude. Accordingly, it is meaningless to compare the minimum hazard magnitudes across hazard types based on the data points plotted on Fig. 4. Considering this, we only focus on comparisons of events with large hazard magnitudes.

Within the datasets for this study, all 37 events with the largest equivalent hazard magnitudes are either a tsunami or a drought,

with their equivalent hazard magnitudes ranging [6.50, 10.21]. The event with the largest equivalent hazard magnitude is the 1960 Chilean tsunami that killed 6 thousand and affected over 2 million population in Chile as well as resulted in 61 fatalities in Hawaii, USA. The well-known 2004 Indian Ocean tsunami that killed more than 2 million people ranks 10th among all events, with its equivalent hazard magnitude at 8.27. The drought event with the largest equivalent hazard magnitude (9.07) is the 2002 Indian monsoon drought that affected a total of about 300 million people. The largest earthquake events, with

equivalent hazard magnitude at 6.41, include the 1920 Haiyuan earthquake in mainland China that resulted in at least 180 thousand fatalities. Among the considered 12 hazard types, the natural hazard with the lowest maximum equivalent hazard magnitude is tornado. The tornado event with the largest equivalent hazard magnitude (3.62) is the 2013 El Reno tornado in Oklahoma, USA. This tornado event led to a total damage of about 3.4 billion 2019 USD (Guha-Sapir et al. 2021).

### 4.2.1 Earthquake, Tornado, Forest Fire, and Tropical Cyclone

Figure 5 compares hazard magnitudes of events of four hazard types, i.e., earthquake, tornado, forest fire, and tropical cyclone, with ranges of hazard magnitudes adjusted according to the earthquake Richter magnitude scale. The figure shows that tornadoes tend to have a smaller hazard magnitude than large earthquakes and tropical cyclones. Most of the recorded tornadoes have a hazard magnitude equivalent to an earthquake Richter magnitude between 5 and 6. Compared with tropical cyclones in terms of peak sustained wind speed on the Saffir–Simpson hurricane wind scale, these tornadoes only have the size of a tropical

storm. This result indicates that hazard strength of an entire tornado event may be much smaller than the one for a large earthquake or tropical cyclone, even though tornadoes can still cause significant damage locally as in the case of the 2013 El Reno tornado. Meanwhile, the wide spread of data points of tornadoes with respect to hazard magnitude on Fig. 5a suggests that exposed value and susceptibility of exposed entities may be much stronger predictors of hazard impacts than hazard magnitude for tornado events.



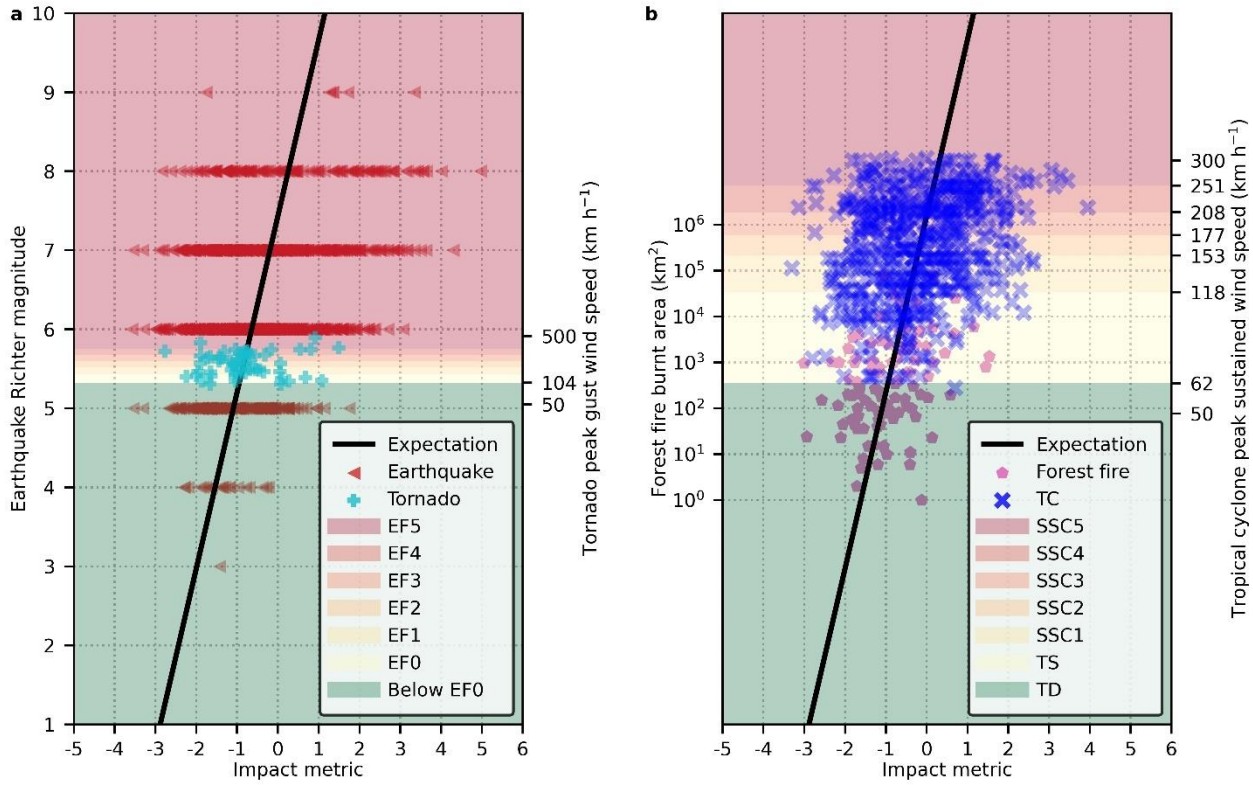


**Figure 5: Comparisons of hazard magnitudes of four hazard types. (a)** Earthquake Richter magnitude versus tornado enhanced Fujita scale. EF0, EF1, EF2, EF3, EF4, and EF5: enhanced Fujita scale 0, 1, 2, 3, 4, and 5 with gust wind speed at 104–137, 138–177, 178–217, 218–266, 267–322, and over 322 km h$^{-1}$, respectively. **(b)** Forest fire burnt area versus tropical cyclone Saffir–Simpson wind scale. TD and TS: tropical depression and tropical storm with sustained wind speed below 63 km h$^{-1}$ and at 63–118 km h$^{-1}$, respectively; SSC1, SSC2,
SSC3, SSC4, and SSC5: Saffir–Simpson category 1, 2, 3, 4, and 5 with sustained wind speed at 119–153, 154–177, 178–208, 209–251, and over 251 km h$^{-1}$, respectively; TC: tropical cyclone. **(a)** and **(b)** are plotted with the same range and scale with respect to the earthquake Richter magnitude.





Compared to earthquakes, tropical cyclones that reach a hurricane level on the Saffir–Simpson scale are equivalent in hazard magnitude to an earthquake with a Richter magnitude greater than 6.5. A magnitude 8 earthquake on the Richter scale has a

similar size in hazard magnitude as a tropical cyclone labelled with a peak category 5 on the Saffir–Simpson scale. Within the datasets for this study, Typhoon Meranti is the tropical cyclone with the largest equivalent hazard magnitude at 5.66. Although the typhoon was strong and affected the Philippines, Taiwan, mainland China, and South Korea in September 2016, it only resulted in a total economic loss of around 70 million 2019 USD, according to the EM-DAT database (Guha-Sapir et al. 2021). In addition to earthquake and tropical cyclone, forest fire is another hazard type with a statistically significant estimate of

coefficient of hazard magnitude indicator (Table 4). However, forest fires tend to have smaller equivalent magnitudes than large earthquakes and tropical cyclones (Fig. 4b). The two largest forest fires within the dataset had an equivalent hazard magnitude of 4.33. They occurred in Russia and Mongolia in 1996, resulting in 19 and 25 fatalities, respectively (Guha-Sapir et al. 2021). Both forest fires are equivalent to a tropical cyclone with its peak sustained wind speed reaching category 1 on the Saffir–Simpson scale. They are also equivalent in hazard magnitude to an earthquake with a Richter magnitude between

6.5 and 7.

### 4.2.2 Cold Wave and Heat Wave

With Fig. 6, we can compare the hazard magnitudes of cold wave and heat wave events. Both hazard types have a narrow range of equivalent hazard magnitude of hazard events, with [4.54, 5.79] for cold wave and [4.79, 5.67] for heat wave (also see supplementary material Data S5). This is also consistent with the statistically insignificant estimates of their corresponding

coefficients of hazard magnitude indicators (Table 4). Despite the narrow ranges of equivalent hazard magnitude, the range of minimum temperature of cold wave events from 0 °C to –55 °C is approximately equivalent to the range of maximum temperature of heat wave events from 30 °C to 55 °C (Fig. 6). The strongest cold wave event recorded in the data occurred in Russia in 2001, with its minimum temperature at –57 °C. This cold wave event killed 145 people, affected 6 120 more, and led to an economic loss of 100 thousand 2019 USD. On the other hand, the heat wave event with the largest hazard magnitude

had a maximum temperature at 53 °C. It struck Pakistan in June 1991, resulting in 523 human fatalities (Guha-Sapir et al., 2021).




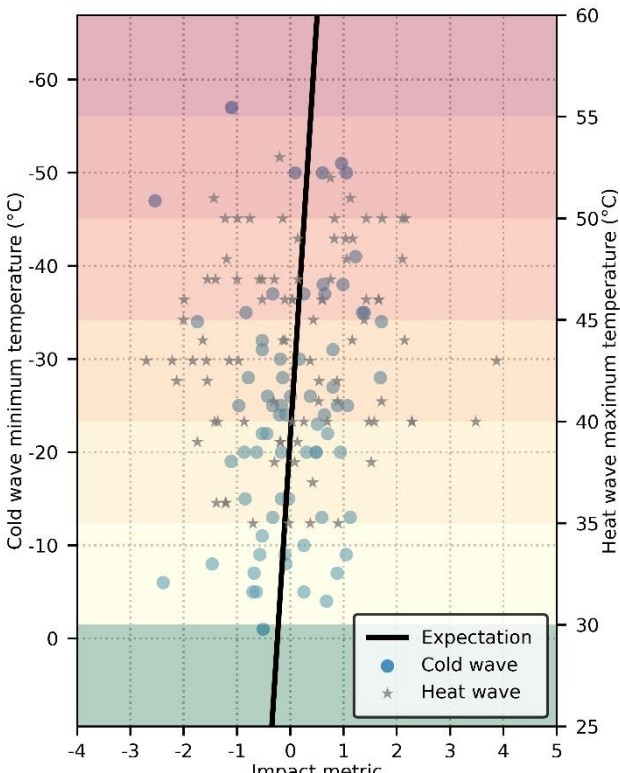

Figure 6: Cold wave minimum temperature versus heat wave maximum temperature.



### 4.2.3 Riverine Flood and Drought

Another comparison of hazard magnitude can be conducted between riverine flood and drought events (Fig. 7). Among hazard events included in the datasets for this study, drought has a large range of equivalent hazard magnitude of [3.23, 9.07], while riverine flood has a relatively small range of [2.11, 5.59]. A riverine flood event with a flooded area of 100 $km^2$ is equivalent in hazard magnitude to a drought event with an affected area of about 1 $km^2$. Meanwhile, a drought event with an affected area of 100 $km^2$ has the similar hazard magnitude as a riverine flood with a flooded area of 1 million $km^2$. Here, because the

magnitude indicators of riverine flood and drought are defined by the EM-DAT database without strong justifications (Guha-Sapir et al. 2021), the meanings and modelling of the presented magnitude indicators of these two hazard types may deserve further investigation. Nevertheless, large drought events seem to be much larger in hazard magnitude than large riverine floods, even though some riverine floods may lead to more severe impacts. For example, the riverine flood event in mainland China in 1998 has an equivalent hazard magnitude of 4.99. But the event resulted in over 3 600 fatalities, more than 238 million

affected population, and an economic loss of 30 billion 2019 USD (Guha-Sapir et al. 2021).

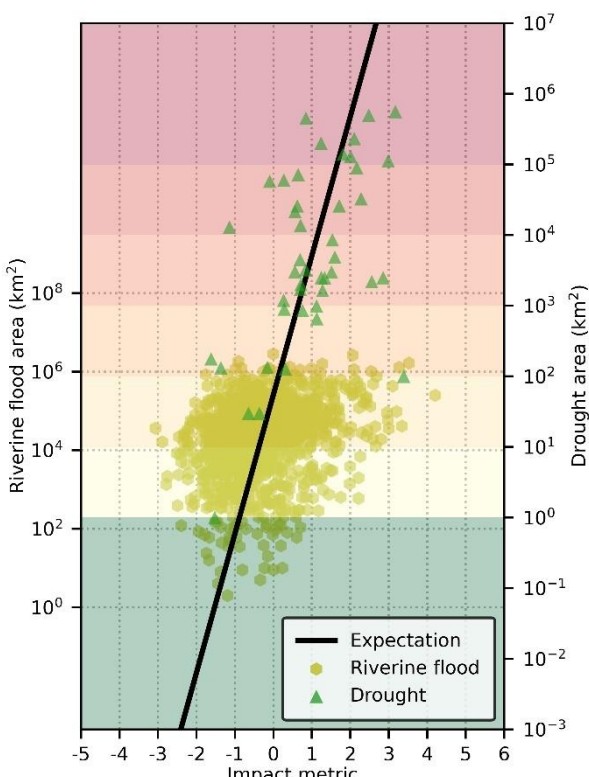

**Figure 7: Riverine flood area versus drought area.**





## 4.3 Sensitivity Analysis

In this study, the impact metric was constructed as the principal component of three transformed impact variables. The sum of
squares of weights of transformed impact variables within the impact metric equals one. We conducted a visual sensitivity analysis to examine if alterations of weights of transformed impact variables within the impact metric can significantly affect the relative comparison of hazard magnitudes across hazard types. For this sensitivity analysis, we first kept the sum of squares of all weights of transformed impact variables equal to one. Second, we maintained an equal ratio of squares of weights between two transformed impact variables. Third, we changed the weight of the third transformed impact variable and adjusted
the weights of the other two transformed impact variables according to the first two rules.

Figure 8 shows the result of a sensitivity analysis with data points of tsunami and flash flood as a demonstrative example. Data points are plotted based on their equivalent hazard magnitudes with a fixed scale of the hazard magnitude indicator of tsunami. When the weight of each of the transformed impact variables of fatality (Figs. 8a–8d), total affected population (Figs. 8e–8h), and total economic damage (Figs. 8i–8l) is shifted from zero to one, there are identifiable increasing or decreasing trends of
alterations of the distributions of data points as well as the deviations between clusters of data points of the two different hazard types. However, when weights of transformed impact variables are far away from the extreme value of zero or one, there is no significant change regarding the distribution of data points with respect to equivalent hazard magnitude (see Figs. 6b, 6c, 6f, 6g, 6j, and 6k). This result indicates desirable performance of the proposed methodology for deriving equivalent hazard magnitude of an event on the Gardoni Scale.




**Figure 8: Results of visual sensitivity analysis regarding effects of altering weight of one transformed impact variable within impact metric on equivalent magnitudes of tsunami and flash flood events.** (a) Fatality weight equals zero. (b) Fatality weight equals $\sqrt{w_F^2/2}$, where $w_F$ is the weight regarding fatality. (c) Fatality weight equals $\sqrt{(w_F^2+1)/2}$. (d) Fatality weight equals one. (e) Total affected population weight equals zero. (f) Total affected population weight equals $\sqrt{w_{TA}^2/2}$, where $w_{TA}$ is the weight regarding total affected population. (g)



Total affected population weight equals $\sqrt{(w_{\text{TA}}^2 + 1)/2}$. **(h)** Total affected population weight equals one. **(i)** Total damage weight equals zero.

**(j)** Total damage weight equals $\sqrt{w_{\text{TD}}^2/2}$, where $w_{\text{TD}}$ is the weight regarding total damage. **(k)** Total damage weight equals $\sqrt{(w_{\text{TD}}^2 + 1)/2}$. **(l)**

Total damage weight equals one. In **(a)**–**(l)**, sum of squared weights of three transformed impact variables equals one and the ratio of squares of the other two variable weights were kept constant.



## 5 Discussion

**5.1 Contributions**

To our knowledge, this study represents the first attempt to produce an equivalent hazard magnitude scale, i.e., the Gardoni Scale, to quantify agential-durational hazard strengths for hazard events across multiple hazard types. The proposed scale has several merits. First, professionals in natural hazard and emergency management could use equivalent hazard magnitudes on the Gardoni Scale to facilitate hazard communication among various stakeholders. Similarly, journalists and news media could

adopt the Gardoni Scale for news reporting on natural disasters to the public. When events of different hazard types are described as equivalent to each other in terms of their natural forces, we can use the proposed methodology to compute the equivalent hazard magnitudes of these events on the Gardoni Scale to confirm such equivalency. For example, if we adopt the minimum temperature of –26 °C at Oklahoma City as the hazard magnitude indicator of the February 2021 cold wave event that severely affected the southern states of USA (Doss-Gollin et al., 2021), we find that the event has an equivalent hazard

magnitude of 5.10 on the Gardoni Scale. This is equivalent to the hazard magnitude of Hurricane Harvey (2017), which had a peak sustained wind speed of 215 km h$^{-1}$, and a Richter magnitude slightly larger than 7.5. Given such information on equivalency of hazard magnitudes across historical events, individuals or decision makers that may have previously experienced one event may be provided with a better understanding of the human, financial, and material resources that are needed to prepare for a predicted hazard event of similar magnitude.

Beside its utility for emergency management of a hazard event, computation of equivalent hazard strengths of events can enhance hazard profiling and risk analysis within a multi-hazard context. When hazard strengths can be evaluated comparatively across hazard types, we can model hazard frequency and exposure regarding multiple types of hazards simultaneously and create multi-hazard hazard maps. With quantified hazard equivalency, we may also derive loss ratio curves with respect to a uniform equivalent hazard strength measure to indicate the differences in vulnerability and resilience of

individuals, communities, and infrastructures facing hazards across different hazard types. Such multi-hazard quantification of hazard, exposure, vulnerability, and resilience can be integrated to facilitate risk analysis to predict future losses and loss ratios without having to resort to lengthy efforts to develop sophisticated models for each individual hazard types. Thus, management of perceived and engineered risks due to natural hazard events could become much easier by using a hazard equivalency methodology. To achieve such multi-hazard quantifications of risks of natural hazard events, more research is

needed not only to improve the proposed Gardoni Scale for equivalent agential-durational hazard strengths, but also to explore the modelling of hazard equivalency of other types of hazard strengths, in particular, the locational hazard strengths for hazard management at the local level.

**5.2 Limitations and Future Work**

As shown in the previous section, data points in this study can be visualized as centred along the expectation line, albeit with

a large variation (Fig. 4). This implies that the derived equivalent hazard magnitudes may correspond well to the expectation





of hazard impacts but without precision. Such a lack of precision is expected for this study because impacts of hazard events are not only the result of hazard strength but also correlated with environmental, societal, and infrastructural factors that affect the exposed value and susceptibility of exposed entities within a natural hazard context (Fig. 1). Because of the effect of these factors other than hazard strength, however, the mere inclusion of, or the complete exclusion of, data points with a unique bias

toward one direction of these factors will result in biased derivation of equivalent hazard strength metric. To reduce such a bias, in this study, we included all available data points of hazard events throughout the world and from a long period of 1900– 2020. However, there may still be bias due to spatial or temporal concentrations of data points regarding certain hazard types. Future works need to study how to further reduce this potential bias caused by factors of exposed value and susceptibility of exposed entities.

To demonstrate the implementation of the proposed methodology for deriving equivalent hazard magnitudes of events, we only considered one hazard magnitude indicator for each hazard type. For many hazard types, one indicator cannot represent the true hazard magnitude of an event. For example, both wind and precipitation contribute significantly to damages associated with tropical cyclone events (Mudd et al., 2017). Moreover, selection of hazard magnitude indicators in this study was also limited by the adopted datasets. As an example, the earthquake Richter magnitude (Richter, 1935) was the only recorded hazard

magnitude indicator in the datasets of this study. However, because Richter magnitude is easily subject to saturation for large earthquakes, it has become less often referenced than moment magnitude (Kanamori, 1977) for indicating hazard magnitude of an earthquake event. For flood hazards, as another example, there is a lack of established methods to quantify the agential-durational hazard strength metrics. In this study, we followed the EM-DAT database (Guha-Sapir et al. 2021) to use the flooded area as the hazard magnitude indicator for the flood hazards. However, the definition of such flooded area is still vague and

deserves more research. An ideal agential-durational hazard strength metric for a flood event needs to integrate flood intensity measures, such as water depth, flood volume, and flow velocity, over the entire flooded area and duration of the event to correspond to the total energy released by the natural force of the event. More efforts, therefore, are needed to study, select, and quantify the appropriate hazard magnitude indicators for deriving equivalent hazard magnitudes of events on the Gardoni Scale.

Beside hazard magnitude indicators, construction of the impact metric is pivotal to calibration of regression models for derivation of equivalent hazard magnitudes. In this study, we only leveraged three impact variables to apply PCA to derive the impact metric. However, impacts of a hazard event may affect a variety of realms including physical, social, economic, and environmental well-beings (Lindell and Prater, 2003; Gardoni and Murphy, 2010; Alexander, 2013a; Wang et al., 2016; 2020a). To advance methodological development for the proposed Gardoni Scale and quantification of other equivalent hazard

strength metrics, more studies are needed to scrutinize different indicators as hazard impact variables of hazard events and to seek the optimal models to combine hazard impact variables to inform the level of impacts of hazard events for different hazard types.

To support modelling with consideration of hazard magnitude indicators and impact metric, more statistical, machine learning, and other quantitative models can be attempted to establish the mapping between an equivalent hazard magnitude and the





445 expectation of impacts of hazard events. When data on hazard events with little or zero impacts become available for modelling, we may apply zero-inflated techniques or other methods to consider the effect of data points with zero impacts to improve the derivation of equivalent hazard magnitudes of events within a multi-hazard context.

## 6 Conclusion

In this article, we proposed an equivalent hazard magnitude scale, called the Gardoni Scale, to measure the strength of natural
450 force involved in the entirety of a natural hazard event for comparative analysis across different hazard types. A computational methodology based on PCA and regression modelling was introduced and implemented to demonstrate the methodological utility in derivation of the equivalent hazard magnitudes of events for 12 natural hazard types. The proposed equivalent hazard magnitudes of events on the Gardoni Scale are recommended to be adopted for hazard communication by various stakeholders including news media, decision makers, industry professionals, academic personnel, and the public. By applying the proposed
455 Gardoni Scale, we can also help quantitatively separate the natural forces of hazard events from the environmental, societal, and infrastructural factors of hazard impacts to support social scientific and engineering research in natural hazard phenomena with a multi-hazard approach. We anticipate that this study on equivalent hazard magnitude will be extended to comparative modelling of other types of hazard strengths of events in a multi-hazard manner to consolidate the foundations for quantifying and studying hazard exposure, hazard vulnerability, hazard resilience, and other conditions for disaster risk reduction due to
460 natural hazards at both local and global levels.

## Code and Data Availability

Python codes and data that support this study are available at https://doi.org/10.15139/S3/DJV7CR (Wang and Sebastian, 2020).

## Video Supplement

465 Supplementary Video S1 shows the distribution of data points with respect to impact variables and the impact metric.

## Author Contribution

Y.V.W. was responsible for design of the study, data collection, data processing, and coding. Data analysis and drafting and critical review of the manuscript was undertaken by both authors.





## Competing Interests

The authors declare that they have no conflict of interest.

## Acknowledgements

Y.V.W. thanks Professor Paolo Gardoni and Professor Colleen Murphy for inspiring discussions and suggestions.

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
