# Peer review of "Equivalent Hazard Magnitude Scale"

_Natural Hazards and Earth System Sciences, 2021_

## Referee Comment (RC1)

**Equivalent Hazard Magnitude Scale**
Yi Victor Wang, Antonia Sebastian

Referee - John Hillier

I like the idea of this paper, but it needs to be much more clearly written - I am afraid I had to re-read many times to understand the driving purpose, method proposed, and assumptions. It would benefit greatly from being focussed and simplified. A major revision is needed in terms of the text, whilst the underlying work seems mainly robust.

12 hazards are considered.  The innovation is a creating a new standardized measure of impact (*IM*) by combining 3 loss/impact measures from EM-DAT after log transforming and standardizing data each. *IM* is then related (linear regression) to measures of hazard severity (e.g. Richter scale) for each hazard, such that for each hazard event (e.g. $M_w$ = 6.7) a *IM* can be estimated, which is then linearly scaled to fit a range [0,10], called 'equivalent magnitude' *EM*. Finally, on the premise that hazard characteristics of events that appear in EM-DAT are a representative sample of all similar events, and that averaging (via regression) allows all local risk related aspects (e.g. exposed assets, vulnerability) to cancel out, the authors argue that measures of hazard severity (e.g. Richter scale, area flooded) can be compared via their *EM* values. This permits events (e.g. a cat 5 hurricane and a $M_w$ 6.7 earthquake) to be compared in terms of potential to cause damage (i.e. hazard) in a way that is as decoupled as possible from local human exposure (i.e. assets at risk), albeit entirely based upon the relative typical size of impact of each event type.

Please find below some more major comments, and a non-exhaustive selection of minor comments. I have only considered the text in any detail to the end of Section 4.1 as I assume a second round of review will be necessary.

**Major comments**

1. I have substantial difficulty with the authors' desire to name a scale they '*propose*' (L10) in this paper after a person (i.e. Gardoni). This is primarily for two reasons.
   - The first reason is the appropriateness of doing this, something not related to the scientific content of the article, so I explicitly ask the journal's editorial team to take a view. For instance, has Gardoni been asked? Is it in the editor's view acceptable scientific practice?
   - The second reason follows from this, and in my view needs the manuscript (e.g. Abstract, Introduction to be rephrased). If the authors use 'the Gardoni scale', a citation to the work it was developed in is sufficient, without further elaboration. If the scale is developed in this paper, I question the justification for the naming. '*Equivalent hazard*' scale should be sufficient if it's novel ..... and others may call it the Wang Scale later if they so choose.

2. Clarity of writing: Throughout, the paper would benefit from simplification and focussing on key points. Illustratively, L11-21 of the abstract provide details, but make little sense before a detailed reading of the paper. Please seek to provide an overview of purpose and a sense of some of the assumptions involved. To simplify, please consider what is truly

necessary for the paper; e.g. (i) reduce Section 2 to Fig. 1 and a short paragraph (ii) Section 3 could be written considerably more succinctly. And, is Table 3 really need to understand the paper's main point? (iii) Section 3.3 might be best in an Appendix to preserve the flow of the paper.

3. Introduction and framing: This work does something new I think, but the way it is presented does not help this argument.

**Selected minor comments**

L17 - 'we argue' instead of 'we show', you are suggesting something, not providing a definitive and unique answer.

L25 - Use 'hazardous events' rather than 'hazard event'.

L25 - Suggest delete 'with a strong natural force' - example of words that are vague and as such add little meaning and detract from the focus of the text.  I illustrate in the next comment.

L27 - "..... these events. The impact of events, whatever their type, can be quantified directly (e.g. by financial loss )(Hillier *et al*, 2015). Various impact scales have also been proposed including the Bradford ......" - I would just name 1 or 2 scales and put the references at the end of the sentence.

L30-38 - Consider using examples to communicate more clearly e.g. the Christchruch quake in New Zealand is an example of a small quake causing lots of damage.

L53-61 - This paragraph finishing the framing of the work needs re-writing. My first point is observation, and my second is a suggestion.
- I didn't use the Gardoni scale in Hillier *et al* (2015, 2020a). Indeed, how could I have as it is proposed here. In 2015 & 2020a I used financial impact as a metric to allow comparison of multiple hazards and their severity (4 and 7 hazard respectively).  In Hillier *et al* (2020b), I use what I refer to as 'impact-based proxies' for hazard to map and understand the estimated combined severity of two hazards (extreme wind and flooding).
- The work proposed here certainly builds on the limited (i.e. two hazard) work in Hillier (2020b), which itself builds on a substantial history of what I dubbed '*impact-based proxies*' (i.e. hazard measures designed to - hopefully - closely relate to impacts) e.g. *v*3 over a threshold is very established for wind (e.g. refs [33-38] in Hillier 2020b - Southern (1979), Klawa (2003)). So, I suggest starting the paragraph with this context (and likely references for other hazards) building to the necessity of a generalized Equivalent Hazard Scale - perhaps with a structure similar to the bullets below.
  - Impacts (e.g. financial losses) have directly used to compare and understand dependencies between multiple (up to 4 or 7) hazards (e.g. Hillier *et al* 2015, 2020b), but strictly this limits understanding to a particular stakeholder (e.g.

insurers, the UK rail network). Indeed, insurers are very experienced at using loss as a metric to understand the relative significance of various hazards [see detail below].

- o Similar about nuclear sector, perhaps mentioning scenarios [I know this exists, but don't have details to hand].
- o There are also indices that integrate multiple weather extremes, but ...... [again see below].
- o A calibration of hazard to impact has been used to create 'impact-based proxies' for hazard, linking two extremes and allowing them to be studied in a way that is relevant to risk and yet decoupled from the detail of local human exposure (Hillier, 2020a).
- o But, there is not as yet a general multi-hazard measure that permits events (e.g. a cat 5 hurricane and a $M_w$ 6.7 earthquake) to be compared in terms of potential to cause damage (i.e. hazard) in a way that is as decoupled as possible from local human exposure (i.e. assets at risk). And, Hillier (2020a) do not create a scale for ease of comparison. We propose ..........

- Indices of Climate Change for the United States - Karl (1996) Bull Am Met Soc.
- "*The Extreme Climate Index (ECI) is an objective, multi-hazard index ..... of extreme weather events*" Malherbe, J. et al. 2018. The Extreme Climate Index (ECI), a tool for monitoring regional extreme events. In: Climate Change and Adaptive Land Management in southern Africa: Assessments, Changes, Challenges, and Solutions, pp. 144-145

- The need to combine risks (between geographic regions and types of risk) has a greater history than currently acknowledged. '*Accumulation*', '*roll-up*' or '*aggregation*' e.g. see Ch 2.7 of Mitchell-Wallace 'Natural Catastrophe Risk Management and Modelling' for an introduction to this subject (p97-105), and how it has been handled for decades (if not centuries) in the provision of insurance. Very well established commercial products have existed for at least 13 years (e.g. Remetrica/Igloo) i.e. this is my personal memory only from when I first saw then embedded within insurers.

L56 - Gardoni (2014) is very explicitly a *risk* scale, not a hazard scale as proposed here. Please use only references that are directly relevant.

L58 - This manuscript should not depend upon Wang & Sebastian (2021b), so please remove as this is still under review.

L124 - (i) consider splitting section 3.1 into 'Data' and 'Magnitude Indicator', (ii) A few sentences before section 3.1 explaining the overall structure of the Methods would help, similar to my second paragraph in this review.

L127 - Are you sure there are no biases (e.g. omissions) in EM-DAT?

L132 - Which did you keep for each hazard, and why? Please justify choices, providing appropriate references.

L135 & L138 - What duration of gust? (e.g. 3 sec or 10 sec, and at what height). These are important distinctions e.g. for tropical cyclones the recording method and therefore apparent severity differ between the USA and Japan.

L140-L144 - Please justify the thresholds used (e.g. Richter magnitude >= 6).

L146 - Sentence does not make sense. No transformation is needed to fit losses in the range ±infinity. Is the purpose to centre the impact metric on zero?

L145-150 - Please add rationale (i.e. systematic logic for when transformation was needed and when it wasn't).

L198 - by 'by applying' I assume you mean a simulation of individual values, rather than using an expectation from the trend line. Using an expectation would not replicate the variability of the data. Please clarify.

L212 - Section 3.4 & Table 4.  Whilst significance of individual parameters is interesting, please compute and provide $p$ values for the models as a whole, and consider omiting any hazards where the statistical model is not significant.

L270 - Fig. 3 - Are these relationships (i.e. $R^2$ values) all statistically significant? If not, please consider the validity of including them in the paper. Those omitted can simply be removed, helping brevity.

L435 - A fundamental limitation (but also benefit) of any impact-based measure of hazard is that it is specific to a user (i.e. the subject of the potential loss).  The authors have endeavoured to define a widely relevant measure, but a brief discussion of the benefits and limitations of this specific is necessary.

**References**

1. **Hillier, J. K.** and Dixon, R. S. (2020b) Seasonal impact-based mapping of compound hazards *Env. Res. Lett.*, **15,** 114013 doi:10.1088/1748-9326/abbc3d

2. **Hillier, J. K.** , Matthews, T., Wilby, R., Murphy, C. (2020a) Multi-hazard dependencies can increase or decrease risk *Nature Climate Change*, **10**, 595–598 doi:10.1038/s41558-020-0832-y [https://rdcu.be/b5kuz]

3. **Hillier, J. K.,** Macdonald, N., Leckebusch, G., Stavrinides (2015) Interactions between apparently primary weather-driven hazards and their cost. *Env. Res. Lett.* **10**(10), 104003, doi:10.1088/1748-9326/10/10/104003

---

## Author Response (AR1)

November 11, 2021

Editor Natural Hazards and Earth System Sciences (NHESS)

Subject: Resubmission of revised article "Equivalent Hazard Magnitude Scale" with manuscript number nhess-2021-87.

Dear editor,

We thank you and the referees for your careful review of our manuscript (nhess-2021-87) entitled "Equivalent Hazard Magnitude Scale". To address the comments from the review team, we have made major revisions to the manuscript. The revised manuscript has 9 605 words, seventy-one references, eight figures, two tables in the main text, two tables in the appendix, six supplementary data files, and one supplementary video.

A detailed account of how we addressed the comments from the referees is attached below this response letter in a point-by-point style. The changes to the manuscript are summarized as follows:

1) We have modified the abstract to focus on key points of the manuscript.

2) We have significantly revised the introduction section.

3) We have significantly reduced the length of section 2: A Problem of Scales.

4) We have modified the methodology section to make it more succinct and have moved the content and tables on missing values to the appendix.

5) We have modified the results section and the discussion section.

6) We have updated the references accordingly.

7) We have double-checked the event records discussed in the manuscript and have corrected two errors.

8) We have updated Fig. 1 and have modified the figure captions.

9) We have also slightly modified the conclusion section.

We look forward to hearing back from you regarding our revision.

Sincerely,

Yi Victor Wang, Ph.D. Postdoctoral Fellow Center of Excellence in Earth Systems, Modeling and Observations Chapman University ywang2@chapman.edu

Antonia Sebastian, Ph.D. Assistant Professor Department of Earth, Marine and Environmental Sciences University of North Carolina at Chapel Hill asebastian@unc.edu

**Referee – John Hillier**

We thank you very much for your constructive comments and insightful suggestions. In the following, we copy your comments in *italics* and follow with our response. The changes to the manuscript are summarized as follows:

1) We have modified the abstract to focus on key points of the manuscript.

2) We have significantly revised the introduction section.

3) We have significantly reduced the length of section 2: A Problem of Scales.

4) We have modified the methodology section to make it more succinct and have moved the content and tables on missing values to the appendix.

5) We have modified the results section and the discussion section.

6) We have updated the references accordingly.

7) We have double-checked the event records discussed in the manuscript and have corrected two errors.

8) We have updated Fig. 1 and have modified the figure captions.

9) We have also slightly modified the conclusion section.

**Comment:** I like the idea of this paper, but it needs to be much more clearly written -I am afraid I had to re-read many times to understand the driving purpose, method proposed, and assumptions. It would benefit greatly from being focussed and simplified. A major revision is needed in terms of the text, whilst the underlying work seems mainly robust.

**Response:** Thank you very much for your encouragement. We have made substantial revisions to the manuscript following your comments and suggestions.

**Comment:** 12 hazards are considered. The innovation is a creating a new standardized measure of impact (IM) by combining 3 loss/impact measures from EM-DAT after log transforming and standardizing data each. IM is then related (linear regression) to measures of hazard severity (e.g. Richter scale) for each hazard, such that for each hazard event (e.g.  $M_w = 6.7$ ) a IM can be estimated, which is then linearly scaled to fit a range [0,10], called 'equivalent magnitude' EM. Finally, on the premise that hazard characteristics of events that appear in EM-DAT are a representative sample of all similar events, and that averaging (via regression) allows all local risk related aspects (e.g. Richter scale, area flooded) can be compared via their EM values. This permits events (e.g. a cat 5 hurricane and a  $M_w$  6.7 earthquake) to be compared in terms of potential to cause damage (i.e. hazard) in a way that is as decoupled as possible from local human exposure (i.e. assets at risk), albeit entirely based upon the relative typical size of impact of each event type.

**Response:** Thank you very much for your summary.

**Comment:** Please find below more major comments, and a non-exhaustive selection of minor comments. I have only considered the text in any detail to the end of Section 4.1 as I assume a second round of review will be necessary.

**Response:** Thank you very much for your comment.

**Comment:** 1. I have substantial difficulty with the authors' desire to name a scale they 'propose' (L10) in this paper after a person (i.e. Gardoni). This is primarily for two reasons. 1.1. The first reason is the appropriateness of doing this, something not related to the scientific content of the article, so I explicitly ask the journal's editorial team to take a view. For instance, has Gardoni been asked? Is it in the editor's view acceptable scientific practice?

**Response:** Thank you very much for your comment and questions. We agree that the final decision about the appropriateness of naming the scale be left to the editorial team. Prior to doing so, we would like to provide some background information regarding this manuscript to address your concern. This manuscript with NHESS is one of a pair of papers recently submitted for peerreviewed journal publication on the topic of equivalent hazard magnitude/intensity. As described in detail in the other manuscript, we have identified four types of equivalent hazard magnitude/intensity: type 1 (agential-durational), type 2 (locational-durational), type 3 (agentialmomental), and type 4 (locational-momental). To differentiate between the equivalent hazard strength scales (types 1 and 2) more easily, YVW would like to name the first two of the scales after his two doctoral co-advisors, Prof. Paolo Gardoni (PG) and Prof. Colleen Murphy (CM), since the original idea of equivalent hazard magnitude/intensity emerged during a doctoral advisory meeting in 2015 with PG and CM in PG's office at the University of Illinois at Urbana-Champaign. Both PG and CM have been approached about this idea and are in accordance with the naming convention. In fact, the type 2 scale (locational-durational) will be named after CM and is already under review after the second round of revision. In this regard, it would seem to be appropriate to name the type 1 equivalent hazard strength scale after PG.

**Comment:** 1.2. The second reason follows from this, and in my view needs the manuscript (e.g. Abstract, introduction to be rephrased). If the authors use 'the Gardoni scale', a citation to the work it was developed in is sufficient, without further elaboration. If the scale is developed in this paper, I question the justification for the naming. 'Equivalent hazard' scale should be sufficient if it's novel ..... and others may call it the Wang Scale later if they so choose.

**Response:** Thank you very much for your comment. During personal communications, PG insisted that we should mention "the Gardoni Scale" in its current manner in the manuscript with its first appearance in the abstract. The scale is developed in this paper. The justification for the naming has been provided within our previous response to the reviewer's comment. Since there can be four different equivalent hazard magnitude/intensity scales, merely using the term "equivalent hazard scale" does not seem to be sufficient. What others may call these scales is beyond the scope of this study. Nevertheless, YVW insists the type 1 equivalent hazard strength scale be called the Gardoni Scale. Such a naming system has already appeared in a recent academic/professional presentation at the 2021 EGU General Assembly (see Wang and Sebastian 2021 at https://doi.org/10.5194/egusphere-egu21-6468).

**Comment:** 2. Clarity of writing: Throughout, the paper would benefit from simplification and focussing on key points. Illustratively, L11-21 of the abstract provide details, but make little sense before a detailed reading of the paper. Please seek to provide an overview of purpose and a sense of some of the assumptions involved.

**Response:** Thank you very much for your comment. Following you suggestion, we have significantly modified our abstract.

The modified abstract now reads: "Hazard magnitude scales are widely adopted to facilitate communication regarding hazard events and the corresponding decision making for emergency management. A hazard magnitude scale measures the strength of a hazard event considering the natural forcing phenomena and the severity of the event with respect to average entities at risk. However, existing hazard magnitude scales cannot be easily adapted for comparative analysis across different hazard types. Here, we propose an equivalent hazard magnitude scale to measure the hazard strength of an event across multiple types of hazards. We name the scale the Gardoni Scale after Professor Paolo Gardoni. We design the equivalent hazard magnitude on the Gardoni Scale as a linear transformation of the expectation of a general measure of adverse impact of a hazard event given average exposed value and vulnerability. With records of 12 hazard types from 1900 to 2020, we demonstrate that the equivalent magnitude can be empirically derived with historical data on hazard magnitude indicators and impacts of events. In this study, we model the impact metric as a function of fatalities, total affected population, and total economic damage. We show that hazard magnitudes of events can be evaluated and compared across hazard types. We find that tsunami and drought events tend to have large hazard magnitudes, while tornadoes are relatively small in terms of hazard magnitude. In addition, we demonstrate that the scale can be used to determine hazard equivalency of individual historical events. For example, we compute that the hazard magnitude of the February 2021 North American cold wave event affecting the southern states of the United States of America was equivalent to the hazard magnitude of Hurricane Harvey in 2017 or a magnitude 7.5 earthquake. Future work will expand the current study in hazard equivalency to modelling of local intensities of hazard events and hazard conditions within a multi-hazard context." (L8-24)

**Comment:** 2.1. To simplify, please consider what is truly necessary for the paper; e.g. (i) reduce Section 2 to Fig.1 and a short paragraph.**

**Response:** Thank you very much for your comment regarding Section 2. We have reduced its length from 919 words to 420 words. However, because this section offers the theoretical background for the proposal of the Gardoni Scale, we have kept some of the prior content. Such a theoretical background would be, for the first time, introduced in a peer-reviewed journal article if accepted earlier than the other submission on the Murphy Scale. As such, it is essential to lay out the four types of hazard strength metrics before introducing the details of the methodology to derive the equivalent hazard magnitude on the proposed Gardoni Scale. Moreover, there are also some fundamental confusion associated with the terminology in the field of disaster studies that need to be clarified before proposing the Gardoni Scale. In light of these, we feel strongly to keep Section 2 within the manuscript.

The modified Section 2 now reads:

"In natural hazards research, theoretical frameworks are often based on basic concepts, such as hazard, impact, exposure, vulnerability, recovery, and resilience, that have overlapping or

discipline-specific definitions (see, e.g., Klijn et al., 2015). These inconsistencies across disciplines often result in confusion in quantitative modelling. Therefore, we clarify several definitions used in this paper. Herein, the impacts of an event are the result of strength of the hazard agent, value of entities exposed to the event, and vulnerability of the exposed entities to hazard impacts (Nigg and Mileti, 1997; Coburn and Spence, 2002; Wisner et al., 2004; Dilley et al., 2005; McEntire, 2005; Adger, 2006; Peduzzi et al., 2009; Burton, 2010; Lindell, 2013; Birkmann et al., 2014; Highfield et al., 2014; van de Lindt et al., 2020; Wang et al., 2020; Wang and Sebastian, 2021a). As shown in Fig. 1, hazard strength of an event is one of the main drivers, albeit not the sole driver, of impacts.

Figure 1: Hazard event impacts as the result of hazard strength, exposed value, and vulnerability of exposed entities.

Hazard strength is often referred to as the hazard magnitude or hazard intensity (Blong, 2003; Alexander, 2018). However, these two concepts are not equivalent. Hazard magnitude is a measure of the size of, or the total energy involved in, the entirety of a hazard event (Blong, 2003; Alexander, 2018), whereas hazard intensity is often a measure of the strength of an event with respect to a given location or area and/or a moment or period.

Recently, Wang and Sebastian (2021b) identified two defining dimensions, i.e., the spatial and temporal dimensions, to categorize existing hazard strength scales. These scales can be classified as *agential* or *locational* along the spatial dimension and *durational* or *momental* along the temporal dimension. A hazard strength scale is categorized as *agential* if it indicates the size of an event within its entire spatial range and *locational* if it is given for a set of locations within the spatial range of an event. Likewise, a hazard strength scale is categorized as *durational* when it corresponds to the entire duration of an event and *momental* when it corresponds to a set of moments within the duration of an event. Considering both the spatial and temporal dimensions, hazard strength scales can therefore be categorized into four types, i.e., the *agential-durational scale*, the *locational-durational scale*, the *agential-momental scale*, and the *locational-momental scale*. In this study, we use term "hazard magnitude" to refer to an agential-durational hazard strength of an event." (L79-101)

**Comment:** 2.2. (*ii*) Section 3 could be written considerably more succinctly. And, is Table 3 really need to understand the paper's main point?**

**Response:** Thank you very much for your comment. Also having considered your later comments, we have made a significant modification to Section 3 to make it more succinct. Table 3 has also been moved to Appendix A. Although the previous Tables 3 and 4 (now Tables A1 and A2) are not necessary for understanding the paper's main point, they are important in providing information for reproduction of the results of this study.

The modified Section 3 now reads:

[revised manuscript text omitted]

**Comment:** 2.3. (iii) Section 3.3 might be best in an Appendix to preserve the flow of the paper.

**Response:** Thank you very much for your comment. We have modified Section 3.3 and moved it to Appendix A.

The new Appendix A now reads: "

**Appendix A: Missing Values and Data Aggregation**

Six simple linear regression models and three multiple linear regression models with two independent variables were calibrated with the same data points for derivation of the impact metric. These regression models were created to fill in missing values of impact variables for data points with at most two empty entries among the three impact variables. Within each of these nine linear regression models, the dependent variable is one of the three impact variables. For each of the six simple linear regression models, the independent variable is one of the two impact variables that are not used as the dependent variable. The simple linear regression models have the form

$$IV_1 = a_1 + b_1 IV_2 + \sigma_1 \varepsilon ,$$
(A1)

where  $a_1 = 0$  and  $b_1$  are two model coefficients,  $IV_1$  and  $IV_2$  are two considered transformed and standardized impact variables, and  $\sigma_1$  is the dispersion parameter. The statistics of parameters of these simple linear regression models are shown in Table A1. Per the three multiple linear regression models with two independent variables, the independent variables are the two impact variables other than the one used as the dependent variable. The formula for the multiple linear regression models is

$$IV_1 = a_2 + b_2IV_2 + c_2IV_3 + \sigma_2\varepsilon ,$$
(A2)

where  $a_2 = 0$ ,  $b_2$ , and  $c_2$  are three model coefficients,  $IV_3$  is the third transformed and standardized impact variable, and  $\sigma_2$  is the dispersion parameter. Table A2 lists the statistics of parameters of the multiple linear regression models with two independent variables. The missing values of data points were filled with the expectations regressed on the independent variables with available data. The data were then aggregated event-wise to form data points of the dataset for deriving the equivalent hazard magnitudes.

Table A1: Statistics of parameters of six simple linear regression models for filling in missing values of impact variablesa.

| Model number    | Dependent variable         | Independent variable      | b 1 | $\sigma_1$ |
|-----------------|----------------------------|---------------------------|-----------------------|------------|
| 11              | Fotolity                   | Total affected population | 0.5096                | 0.8604     |
| 11              | Tatanty                    | Total affected population | (0.0224)              | (0.0159)   |
| 12              | Estality                   | Total damage              | 0.2802                | 0.9599     |
| 12              | Fatanty                    | Total damage              | (0.0250)              | (0.0177)   |
| I3 b | Total offected nonvilation | Fatality                  | 0.5096                | 0.8604     |
|                 | Total affected population  |                           | (0.0224)              | (0.0159)   |
| I4 Total a      | Total offected nonvelation | Total damaga              | 0.2948                | 0.9556     |
|                 | Total affected population  | 10tal dallage             | (0.0249)              | (0.0176)   |
| I5°             | Total damage               | Estality                  | 0.2802                | 0.9599     |
|                 |                            | Fatality                  | (0.0250)              | (0.0177)   |
| TCd             | Total domogo               | Total offected completion | 0.2948                | 0.9556     |
| 104             | i otal damage              | Total affected population | (0.0249)              | (0.0176)   |

aThis table corresponds to supplementary material Data S2; R-squared measures are included in Fig. 2; standard errors are in the parentheses; estimations of  $b_1$  and  $\sigma_1$  are all significant at  $p < 10^{-20}$ .

bModels I1 and I3 share the same model parameters and R-squared measures.

cModels I2 and I5 share the same model parameters and R-squared measures.

dModels I4 and I6 share the same model parameters and R-squared measures.

| Table A2: Statistics of parameters of thre        | e multiple linear regression | 1 models with two in | dependent variables | for filling in |
|---------------------------------------------------|------------------------------|----------------------|---------------------|----------------|
| missing values of impact variables a . |                              |                      |                     |                |

| Model
number | Dependent variable | Independent variable
1 | Independent variable 2 | b 2 | c 2 | $\sigma_2$ |
|-----------------|--------------------|---------------------------|------------------------|-----------------------|-----------------------|------------|
| Ι7              | Fatality           | Total affected            | Total damage           | 0.4676                | 0.1423                | 0.8496     |
|                 |                    | population                |                        | (0.0232)              | (0.0232)              | (0.0157)   |
| I8              | Total affected     | Fatality                  | Total damage           | 0.4633                | 0.1650                | 0.8457     |
|                 | population         |                           |                        | (0.0230)              | (0.0230)              | (0.0156)   |
| 19              | Total damage       | Fatality                  | Total affected         | 0.1755                | 0.2054                | 0.9435     |
|                 |                    |                           | population             | (0.0286)              | (0.0286)              | (0.0174)   |

aThis table corresponds to supplementary material Data S3; R-squared measures are included in Fig. 2; standard errors are in the parentheses; estimations of  $b_2$ ,  $c_2$ , and  $\sigma_2$  are all significant at  $p < 10^{-8}$ ." (L412-441)

**Comment:** 3. Introduction and framing: this work does something new I think, but the way it is presented does not help this argument.

**Response:** Thank you very much for your comment. We have reframed the Introduction section and Section 2, also with consideration of your later comments, to improve the introduction and framing of our research work.

The modified Introduction section now reads:

[revised manuscript text omitted]

vulnerability (Hillier et al., 2020). Nevertheless, there is not yet a general metric that permits events of different hazard types to be compared in terms of potential to cause damage in a way that is as decoupled as possible from exposed values and vulnerability.

To enable evaluation of event-wise hazard strengths across different hazard types, in this article, we propose a multi-hazard *equivalent hazard magnitude scale* – the *Gardoni Scale* – for natural hazards. The proposed scale is named after the Alfredo H. Ang Family Professor Paolo Gardoni at the University of Illinois at Urbana–Champaign. Because hazard strength is correlated with hazard impacts given average exposed value and vulnerability of considered entities, the expectation of a metric of observed impacts of hazard events can be used to calibrate models for deriving equivalent hazard magnitudes (Hillier et al., 2015; Hillier and Dixon, 2020; Wang and Sebastian, 2021b). In this article, a quantitative modelling methodology based on a principal component analysis (PCA) and a set of linear regressions is developed to construct the impact metric and derive equivalent hazard magnitudes on the Gardoni Scale. The impact metric is a function of three impact variables, i.e., fatality, total affected population, and total damage in 2019 USD. We use historical event data from the EM-DAT International Disaster Database (Guha-Sapir et al., 2021) from 1900 to 2020 to calibrate the quantitative models. To demonstrate the value of the proposed scale, we apply it to discuss the equivalent magnitudes of historical and recent hazard events.

The subsequent sections are organized as follows. First, we provide a brief theoretical background for this study. We then introduce our methodology, including data processing, to derive the equivalent hazard magnitude on the Gardoni Scale. Next, we lay out the results of applying our methodology and compare natural hazard types regarding the derived equivalent hazard magnitudes. Finally, we discuss the potential contributions and limitations of the proposed scale before concluding the article." (L26-77)

**Comment:** *L17* – 'we argue' instead of 'we show', you are suggesting something, not providing a definitive and unique answer.

**Response:** Thank you very much for your comment. To better present what we wish to convey here, we have changed "we show" into "we compute".

The corresponding sentence now reads: "For example, we compute that the hazard magnitude of the February 2021 North American cold wave event affecting the southern states of the United States of America was equivalent to the hazard magnitude of Hurricane Harvey in 2017 or a magnitude 7.5 earthquake." (L20-22)

**Comment:** *L*25 – *Use 'hazardous events' rather than 'hazard event'.**

**Response:** Thank you very much for your comment. We have modified "hazard events" to "hazardous events" accordingly.

The modified sentence now reads:

"Hazardous events, such as earthquakes, floods, and forest fires, can inflict heavy losses to communities when people and property are exposed to the natural forces of these events." (L28-29)

**Comment:** *L25* – *Suggest delete 'with a strong natural force' – example of words that are vague and as such add little meaning and detract from the focus of the text. I illustrate in the next comment.*

**Response:** Thank you very much for your comment. We have correspondingly deleted 'with a strong natural force'.

The modified sentence now reads:

"Hazardous events, such as earthquakes, floods, and forest fires, can inflict heavy losses to communities when people and property are exposed to the natural forces of these events." (L28-29)

**Comment:** L27 - "..... these events. The impact of events, whatever their type, can be quantified directly (e.g. by financial loss)(Hillier et al, 2015). Various impact scales have also been proposed including the Bradford ....." – I would just name 1 or 2 scales and put the references at the end of the sentence.

**Response:** Thank you very much for your comment. We have modified the sentences accordingly but have chosen to keep the references to all four scales. In particular, we have added a sentence "The impacts of events, whatever their type, can be quantified directly (e.g., by financial loss; Hillier et al., 2015), or estimated on a scale" (L29-30), as suggested.

These modified sentences now read:

"Hazardous events, such as earthquakes, floods, and forest fires, can inflict heavy losses to communities when people and property are exposed to the natural forces of these events. The impacts of events, whatever their type, can be quantified directly (e.g., by financial loss; Hillier et al., 2015), or estimated on a scale. To estimate the impacts of an event with the consideration of its hazard strength, various impact scales have been proposed, including the Bradford disaster scale (Keller et al., 1992; 1997), unified localizable crisis scale (Rohn and Blackmore, 2009; 2015), disaster impact index (Gardoni and Murphy, 2010), and cascading disaster magnitude (Alexander, 2018)." (L28-33)

**Comment:** L30-38 – Consider using examples to communicate more clearly e.g. the Christchurch quake in New Zealand is an example of a small quake causing lots of damage.

**Response:** Thank you very much for your comment. We have incorporated your suggestion on providing examples in the revised version of the manuscript.

The modified sentences now read:

"However, a hazard strength scale is not the same as a hazard impact scale, as impacts are also driven by the exposure and vulnerability of entities, such as individuals, communities, and infrastructures, to an event. This makes it difficult to use impact scales to compare hazard strengths across natural hazard types. For example, the 2011 Christchurch earthquake was one of the most destructive earthquakes in New Zealand, albeit with a medium hazard strength of 6.2 in terms of its moment magnitude (Kaiser et al., 2012). Meanwhile, the 1964 Alaskan earthquake, with a larger moment magnitude of 9.2, resulted in fewer casualties and less economic damage than the Christchurch earthquake (United States Geological Survey [USGS], 2021)." (L33-39)

**Comment:** *L53-61 – This paragraph finishing the framing of the work needs re-writing. My first point is observation, and my second is a suggestion.*

**Response:** Thank you very much for your comment. Also having considered your following comments and suggestions, we have added a new paragraph to enhance the framing of the presented research.

The new paragraph reads:

"To quantify hazard strengths for cross-hazard comparison, impacts can be used to explore dependencies between multiple hazards (e.g., Hillier et al., 2015; Hillier and Dixon, 2020). As an example, insurance professionals often leverage loss metrics to understand the relative significance of various hazards (see, e.g., Mitchell-Wallace et al., 2017). However, their cross-hazard practices of risk aggregation and accumulation are often focused on the exposed values and observed impacts, rather than hazard strengths. In contrast, risk quantification for nuclear facilities requires consideration of hazard strengths across multiple hazard types to facilitate probabilistic safety assessment within a multi-hazard context (see, e.g., Choi et al., 2021). Indices regarding hazard strengths for multiple hazard types have also been created and adopted for extreme meteorological events (see, e.g., Malherbe et al., 2020). When quantifying hazard strengths within a multi-hazard context, a calibration of hazard strength to the expectation of impact may be used to create impactbased proxies for hazard strengths, linking two extremes and allowing them to be studied in a way that is relevant to risk assessment and yet decoupled from the detail of exposed values and vulnerability (Hillier et al., 2020). Nevertheless, there is not yet a general metric that permits events of different hazard types to be compared in terms of potential to cause damage in a way that is as decoupled as possible from exposed values and vulnerability." (L50-62)

**Comment:** I didn't use the Gardoni scale in Hillier et al (2015, 2020a). Indeed, how could I have as it is proposed here. In 2015 & 2020a I used financial impact as a metric to allow comparison of multiple hazards and their severity (4 and 7 hazard respectively). In Hillier et al (2020b), I use what I refer to as 'impact-based proxies' for hazard to map and understand the estimated combined severity of two hazards (extreme wind and flooding).

**Response:** Thank you very much for your comment. We agree that the "impact-based proxies" are a brilliant idea.

**Comment:** The work proposed here certainly builds on the limited (i.e. two hazard) work in Hillier (2020b), which itself builds on a substantial history of what I dubbed 'impact-based proxies' (i.e. hazard measures designed to – hopefully – closely relate to impacts) e.g. v3 over a threshld is very established for wind (e.g. refs [33-38] in Hillier 2020b – Southern (1979), Klawa (2003)). So, I suggest starting the paragraph with this context (and likely references for other hazards) building to the necessity of a generalized Equivalent Hazard Scale – perhaps with a structure similar to the bullets below.

**Response:** Thank you very much for your comment. We have referenced your suggestions and added a paragraph to improve the framing of our presented work, as shown in our response prior to the previous one.

**Comment:** Impacts (e.g. financial losses) have directly used to compare and understand dependencies between multiple (up to 4 or 7) hazards (e.g. Hillier et al 2015, 2020b), but strictly this limits understanding to a particular stakeholder (e.g. insurers, the UK rail network). Indeed, insurers are very experienced at using loss as a metric to understand the relative significance of various hazards [see detail below].

**Response:** Thank you very much for your comment. As shown in one of our responses previously, we have added a paragraph to improve the framing of our presented work. In particular, we now highlight the experiences of insurers in leveraging loss as a common metric for understanding risks.

**Comment:** Similar about nuclear sector, perhaps mentioning scenarios [I know this exists, but don't have details to hand].

**Response:** Thank you very much for your comment. We have also added some content in the new paragraph on the nuclear sector based on some outstanding recent research work on multi-hazard risk assessment for nuclear power plants.

**Comment:** *There are also indices that integrate multiple weather extremes, but ...... [again see below].*

**Response:** Thank you very much for your comment. We have also included the material on the weather extremes in the new paragraph shown previously.

**Comment:** A calibration of hazard to impact has been used to create 'impact-based proxies' for hazard, linking two extremes and allowing them to be studied in a way that is relevant to risk and yet decoupled from the detail of local human exposure (Hillier, 2020a).

**Response:** Thank you very much for your comment. This suggestion has been adopted and incorporated into the new paragraph displayed previously.

**Comment:** But, there is not as yet a general multi-hazard measure that permits events (e.g. a cat 5 hurricane and a  $M_w$  6.7 earthquake) to be compared in terms of potential to cause damage (i.e. hazard) in a way that is as decoupled as possible from local human exposure (i.e. assets at risk). And, Hillier (2020a) do not create a scale for ease of comparison. We propose .....

**Response:** Thank you very much for your comment. We have also adopted this suggestion to develop the new paragraph shown previously.

**Comment:** Indices of Climate Change for the United States – Karl (1996) Bull Am Met Soc. "The Extreme Climate Index (ECI) is an objective, multi-hazard index ..... of extreme weather events" Malherbe, J. et al. 2018. The Extreme Climate Index (ECI), a tool for monitoring regional extreme events. In: Climate Change and Adaptive Land Management in southern Africa: Assessments, Changes, Challenges, and Solutions, pp. 144-145

**Response:** Thank you very much for your comment. The material on meteorological extremes has been added to the new paragraph shown previously.

**Comment:** The need to combine risks (between geographic regions and types of risk) has a greater history than currently acknowledged. 'Accumulation', 'roll-up' or 'aggregation' e.g. see Ch 2.7 of Mitchell-Wallace 'Natural Catastrophe Risk Management and Modelling' for an introduction to this subject (p97-105), and how it has been handled for decades (if not centuries) in the provision of insurance. Very well established commercial products have existed for at least 13 years (e.g. Remetrica/Igloo) i.e. this is my personal memory only from when I first saw then embedded within insurers.

**Response:** Thank you very much for your comment. We have integrated material on insurance practices into the new paragraph shown previously. In our new paragraph, we also emphasize that "As an example, insurance professionals often leverage loss metrics to understand the relative significance of various hazards (see, e.g., Mitchell-Wallace et al., 2017). However, their cross-hazard practices of risk aggregation and accumulation are often focused on the exposed values and observed impacts, rather than hazard strengths." (L51-54)

**Comment:** *L56 – Gardoni (2014) is very explicitly a risk scale, not a hazard scale as proposed here. Please use only references that are directly relevant.*

**Response:** Thank you very much for your comment. We have removed the unnecessary references.

The modified sentence now reads: "The proposed scale is named after the Alfredo H. Ang Family Professor Paolo Gardoni at the University of Illinois at Urbana–Champaign." (L64-65)

**Comment:** *L58 – This manuscript should not depend upon Wang & Sebastian (2021b), so please remove as this is still under review.*

**Response:** Thank you very much for your comment. We have replaced the submitted manuscript under review with a presentation at the 2021 EGU General Assembly available at https://doi.org/10.5194/egusphere-egu21-6468.

**Comment:** L124 – (i) consider splitting section 3.1 into 'Data' and 'Magnitude Indicator'**

**Response:** Thank you very much for your comment. We have made modifications to this section and reduced its content. Since the second and third paragraphs of this section are still about data description, we have kept them within the same section. We have also changed the heading of this section into "Data Collection".

The modified two sections now read: "

**3.1 Data Collection**

To reduce the biases in model calibration due to different protocols for data collection across different types of natural hazards, we only used data gathered from the EM-DAT database (Guha-Sapir et al., 2021). To be included in the EM-DAT database, a hazard event must meet at least one of three criteria, i.e., 10 or more human fatalities, 100 or more people affected by the event, or a declaration of a state of emergency or an appeal for international assistance by a country (Guha-Sapir et al., 2021). For this study, we downloaded the entire EM-DAT datasets on all types of natural hazards. However, due to a lack of records of hazard magnitude indicators of events for

some hazard types (e.g., the volcanic activities and landslides), we only included 12 hazard types. The final dataset for deriving the equivalent hazard magnitudes contained a total of 3 844 data points, each representing one unique hazard event.

The 12 considered hazard types, with their corresponding hazard magnitude indicators listed in parentheses, include: 1) cold wave (minimum temperature in °C); 2) convective storm (peak gust wind speed in km h-1); 3) drought (total affected area in km2); 4) earthquake (Richter magnitude); 5) extra-tropical storm (peak gust wind speed in km h-1); 6) flash flood (total flooded area in km2); 7) forest fire (total burnt area in km2); 8) heat wave (maximum temperature in °C); 9) riverine flood (total flooded area in km2); 10) tornado (peak gust wind speed in km h-1); 11) tropical cyclone (maximum sustained wind speed in km h-1); and 12) tsunami (earthquake Richter magnitude). For data quality control, we removed data points with questionable values of hazard magnitude indicators from our datasets. For cold wave events, we only included data points with a minimum temperature  $\leq 0$  °C; for convective storms, we only considered data points with a burnt area  $\leq 200$  thousand km2; for heat wave events, we only included data points with a burnt area  $\leq 200$  thousand km2; for heat wave events, we only included data points with a burnt area  $\leq 200$  thousand km2; for tornadoes, we only included data points with a maximum temperature  $\geq 35$  °C and  $\leq 57$  °C; for tornadoes, we only included data points with a peak gust wind speed  $\geq 100$  km h-1; and for tsunami, we only considered data points with an earthquake Richter magnitude  $\geq 6$ .

To facilitate regression modelling, we logarithmically transformed values of hazard magnitude indicators to be close to a Gaussian distribution within the range  $(-\infty, \infty)$  for eight of the hazard types. The indicators that were not logarithmically transformed included minimum temperature of cold waves, Richter magnitude of earthquakes, maximum temperature of heat waves, and earthquake Richter magnitude of tsunami. Cold wave and heat wave events were excluded from logarithmic transformations because Celsius temperature has a range  $[-273.15, \infty)$  similar to  $(-\infty, \infty)$ . Meanwhile, the range of an earthquake Richter magnitude is already a desired  $(-\infty, \infty)$ ." (L112-137)

**Comment:** *L124* – (*ii*) *A few sentences before section 3.1 explaining the overall structure of the Methods would help, similar to my second paragraph in this review.*

**Response:** Thank you very much for your comment. We have adopted your suggestion and added a paragraph before Section 3.1.

The new paragraph before Section 3.1 reads: "To quantify hazard strength in terms of equivalent hazard magnitude, we considered 12 hazard types: cold wave, convective storm, drought, earthquake, extra-tropical storm, flash flood, forest fire, heat wave, riverine flood, tornado, tropical cyclone, and tsunami. A general standardized metric of impact was created by combining three loss measures from the EM-DAT database (Guha-Sapir et al., 2021): fatality, total affected population, and total damage. The impact metric was then related to an indicator of hazard strength, such as the Richter magnitude, for each hazard type via linear regression. The expectation of impact metric for each hazard type was linearly scaled and adopted as the equivalent hazard magnitude. Here, two assumptions were made. First, we assumed that the EM-DAT records were not significantly biased across similar hazard events. Second, we assumed that the derivation of expectation of impact metric cancelled out all local factors of exposed value and vulnerability. The following sections outline the method in detail." (L103-111)

**Comment:** *L127 – Are you sure there are no biases (e.g. omissions) in EM-DAT?**

**Response:** Thank you very much for your question. We do not deny that there could be biases in EM-DAT datasets due to omissions of events. However, the main reason why we only used data from EM-DAT is that we tried to reduce the biases due to different protocols for data collection by different databases.

As mentioned in the modified Data Collection section, "To reduce the biases in model calibration due to different protocols for data collection across different types of natural hazards, we only used data gathered from the EM-DAT database (Guha-Sapir et al., 2021)." (L113-114)

**Comment:** *L132* – *Which did you keep for each hazard, and why? Please justify choices, providing appropriate references.**

**Response:** Thank you very much for your comment. The EM-DAT database only includes one magnitude indicator for each natural hazard type. Therefore, we could only select at most one hazard magnitude indicator for each hazard type. The magnitude indicators we kept are, hence, the ones listed in the Data Collection section. To avoid confusion, we have modified the Data section.

The corresponding sentence in the Data Collection section now reads: ". However, due to a lack of records of hazard magnitude indicators of events for some hazard types (e.g., the volcanic activities and landslides), we only included 12 hazard types." (L117-119)

**Comment:** L135&L138 – What duration of gust? (e.g. 3 sec or 10 sec, and at what height). These are important distinctions e.g. for tropical cyclones the recording method and therefore apparent severity differ between the USA and Japan.

**Response:** Thank you very much for your question and comment. Because the EM-DAT database does not provide the details of peak gust wind speeds or peak sustained wind speeds such as seconds and at what height, we were only able to use "peak gust wind speed" and "peak sustained wind speed" to refer to the magnitude indicators for wind-related hazards. We recognize that there is some uncertainty in the data underlying the paper due to the record keeping by EM-DAT and have made a note of our assumptions in the text. For example, we have highlighted the issues with the magnitude indicators of earthquake, flood, and tropical cyclone on L370-380.

These sentences on these issues read: "For example, both wind and precipitation contribute significantly to damages associated with tropical cyclone events (Mudd et al., 2017). Moreover, selection of hazard magnitude indicators in this study was also limited by the adopted datasets. As an example, the earthquake Richter magnitude (Richter, 1935) was the only recorded hazard magnitude indicator in the datasets of this study. However, because Richter magnitude is easily subject to saturation for large earthquakes, it has become less often referenced than moment magnitude (Kanamori, 1977) for indicating hazard magnitude of an earthquake event. For flood hazards, as another example, there is a lack of established methods to quantify the agential-durational hazard strength metrics. In this study, we followed the EM-DAT database (Guha-Sapir et al., 2021) to use the flooded area as the hazard magnitude indicator for the flood hazards. However, the definition of such flooded area is still vague and deserves more research. An ideal agential-durational hazard strength metric for a flood event should integrate multiple flood intensity measures, such as water depth, flood volume, and flow velocity, over the entire flooded

area and duration of the event to correspond to the total energy released by the natural force of the event." (L370-380)

**Comment:** *L140-L144 – Please justify the thresholds used (e.g. Richter magnitude >= 6).**

**Response:** Thank you very much for your comment. These thresholds were set for data quality control to avoid data points with unrealistic values that may have been produced due to human errors. To clarify this point, we have modified the corresponding sentence in the second paragraph of the Data Collection section.

The modified second paragraph of the Data Collection section now reads: "The 12 considered hazard types, with their corresponding hazard magnitude indicators listed in parentheses, include: 1) cold wave (minimum temperature in °C); 2) convective storm (peak gust wind speed in km h-1); 3) drought (total affected area in km2); 4) earthquake (Richter magnitude); 5) extra-tropical storm (peak gust wind speed in km h-1); 6) flash flood (total flooded area in km2); 7) forest fire (total burnt area in km2); 8) heat wave (maximum temperature in °C); 9) riverine flood (total flooded area in km2); 10) tornado (peak gust wind speed in km h-1); 11) tropical cyclone (maximum sustained wind speed in km h-1); and 12) tsunami (earthquake Richter magnitude). For data quality control, we removed data points with questionable values of hazard magnitude indicators from our datasets. For cold wave events, we only included data points with a peak gust wind speed  $\geq$ 60 km h-1; for forest fires, we only included data points with a burnt area  $\leq$ 200 thousand km2; for heat wave events, we only considered data points with a maximum temperature  $\geq$ 35 °C and  $\leq$ 57 °C; for tornadoes, we only included data points with a peak gust wind speed  $\geq$ 100 km h-1; and for tsunami, we only considered data points with a nearthquake Richter magnitude  $\geq$ 6." (L121-131)

**Comment:** L146 – Sentence does not make sense. No transformation is needed to fit losses in the range ±infinity. Is the purpose to centre the impact metric on zero?**

**Response:** Thank you very much for your comment and question. The purpose of logarithmic transformation is to convert hazard magnitude indicators to be close to a Gaussian distribution and to have a range of  $(-\infty, \infty)$  to facilitate regression modeling. Using a transformed magnitude indicator can be more representative across its entire range. To clarify this point, we have modified the last paragraph of the Data Collection section.

The modified last paragraph of the Data Collection section now reads: "To facilitate regression modelling, we logarithmically transformed values of hazard magnitude indicators to be close to a Gaussian distribution within the range  $(-\infty, \infty)$  for eight of the hazard types. The indicators that were not logarithmically transformed included minimum temperature of cold waves, Richter magnitude of earthquakes, maximum temperature of heat waves, and earthquake Richter magnitude of tsunami. Cold wave and heat wave events were excluded from logarithmic transformations because Celsius temperature has a range  $[-273.15, \infty)$  similar to  $(-\infty, \infty)$ . Meanwhile, the range of an earthquake Richter magnitude is already a desired  $(-\infty, \infty)$ ." (L132-137)

**Comment:** *L*145-150 – *Please add rationale (i.e. systematic logic for when transformation was needed and when it wasn't).*

**Response:** Thank you very much for your comment. When the range of a hazard magnitude indicator is or is close to  $(-\infty, \infty)$ , it does not need transformation. To clarify this point, we have modified the last paragraph of the Data Collection section as shown in our previous response.

**Comment:** L198 – by 'by applying' I assume you mean a simulation of individual values, rather than using an expectation from the trend line. Using an expectation would not replicate the variability of the data. Please clarify.

**Response:** Thank you very much for your comment. We actually used the expectations of regression models to fill in the missing values of impact variables. However, this is not a problem for our purpose because the regression models that really matter are the ones showing relationship between hazard magnitude indicator and the impact metric. These regression models for missing values only served to project data points with missing values onto the axis of impact metric. The variation in the impact metric can still be determined by the impact variables without missing values. To clarify that we used the expectation, we have modified the previous Section 3.3 into Appendix A.

The new Appendix A now reads: "Six simple linear regression models and three multiple linear regression models with two independent variables were calibrated with the same data points for derivation of the impact metric. These regression models were created to fill in missing values of impact variables for data points with at most two empty entries among the three impact variables. Within each of these nine linear regression models, the dependent variable is one of the three impact variables for each of the six simple linear regression models, the independent variable is one of the two impact variables that are not used as the dependent variable. The simple linear regression models have the form

$$IV_1 = a_1 + b_1 IV_2 + \sigma_1 \varepsilon ,$$
(A1)

where  $a_1 = 0$  and  $b_1$  are two model coefficients,  $IV_1$  and  $IV_2$  are two considered transformed and standardized impact variables, and  $\sigma_1$  is the dispersion parameter. The statistics of parameters of these simple linear regression models are shown in Table A1. Per the three multiple linear regression models with two independent variables, the independent variables are the two impact variables other than the one used as the dependent variable. The formula for the multiple linear regression models is

$$IV_1 = a_2 + b_2IV_2 + c_2IV_3 + \sigma_2\varepsilon ,$$
(A2)

where  $a_2 = 0$ ,  $b_2$ , and  $c_2$  are three model coefficients,  $IV_3$  is the third transformed and standardized impact variable, and  $\sigma_2$  is the dispersion parameter. Table A2 lists the statistics of parameters of the multiple linear regression models with two independent variables. The missing values of data points were filled with the expectations regressed on the independent variables with available data. The data were then aggregated event-wise to form data points of the dataset for deriving the equivalent hazard magnitudes." (L413-430) **Comment:** *L212* – *Section 3.4 & Table 4. Whilst significance of individual parameters is interesting, please compute and provide p values for the models as a whole, and consider omitting any hazards where the statistical model is not significant.*

**Response:** Thank you very much for your comment. The objective of this manuscript is to propose the Gardoni Scale and to demonstrate how to quantitatively derive the equivalent hazard magnitude on the Gardoni Scale. The purpose of the study is not to reveal the relationship between a hazard magnitude indicator and impact metric, nor is it to predict impact metric with a hazard magnitude indicator. Therefore, we provided the point estimates of section 3.4 and previously Table 4 for reproduction of research results. We chose to include the standard errors and p values to demonstrate that some estimates of model coefficients were statistically significant while others were not. Therefore, we did not omit hazards where the statistical models were not significant.

**Comment:** L270 - Fig. 3 - Are these relationships (i.e.  $R^2$  values) all statistically significant? If not, please consider the validity of including them in the paper. Those omitted can simply be removed, helping brevity.

**Response:** Thank you very much for your comment. Like mentioned in our previous response, the statistical relationships and  $R^2$  values were presented to show that it is okay to have coefficients that are not statistically significant and to have a small  $R^2$  because the purpose of the study is to present a way to compute the equivalent hazard magnitude on the Gardoni Scale. The regression models used in the study are merely tools for computation. Therefore, we have chosen not to omit the regression models that are statistically insignificant. In fact, the significant spread in the data lends insight to whether there could be underlying drivers of impacts such as vulnerability factors that are not considered in studies of hazard equivalency. In the meantime, because of the decoupling from exposure and vulnerability, the derivation of hazard equivalency can provide the benchmark measures of hazard strength to provide a fair foundation for the studies on the effects of those exposure and vulnerability factors across different hazard types.

**Comment:** L435 – A fundamental limitation (but also benefit) of any impact-based measure of hazard is that it is specific to a user (i.e. the subject of the potential loss). The authors have endeavoured to define a widely relevant measure, but a brief discussion of the benefits and limitations of this specific is necessary.

**Response:** Thank you very much for your comment. We have modified the third paragraph of Section 5.2 to incorporate your suggestion regarding the issue of impact metric being specific to a user.

The modified third paragraph of Section 5.2 now reads:

"In addition to hazard magnitude indicators, the construction of the impact metric is important for the calibration of regression models and for the derivation of equivalent hazard magnitudes as it is end-user specific. For example, insurance professionals may be interested in an equivalent hazard magnitude that is derived from data on financial and property loss whereas environmental scientists may be more interested in an impact metric based on ecological damage. Herein, we derived a general metric of impact for equivalent hazard magnitude based on key indicators of societal impact. For this reason, we combined data on fatalities, damages, and affected individuals to derive an impact metric. However, hazard events can affect a variety of sectors resulting in impacts to physical, social, economic, and environmental well-being (Lindell and Prater, 2003; Gardoni and Murphy, 2010; Alexander, 2013; Wang et al., 2016; 2021). To advance methodological development for the proposed Gardoni Scale and quantification of other equivalent hazard strength metrics for various stakeholders, future work should scrutinize different indicators as impact variables of events and to seek the optimal models to combine impact variables to inform the level of impacts of events for different hazard types." (L383-393)

**Anonymous Referee**

We thank you very much for your constructive comments and insightful suggestions. In the following, we copy your comments in *italics* and follow with our response. The changes to the manuscript are summarized as follows:

1) We have modified the abstract to focus on key points of the manuscript.

2) We have significantly revised the introduction section.

3) We have significantly reduced the length of section 2: A Problem of Scales.

4) We have modified the methodology section to make it more succinct and have moved the content and tables on missing values to the appendix.

5) We have modified the results section and the discussion section.

6) We have updated the references accordingly.

7) We have double-checked the event records discussed in the manuscript and have corrected two errors.

8) We have updated Fig. 1 and have modified the figure captions.

9) We have also slightly modified the conclusion section.

**Comment:** The paper introduces a new magnitude scale (the Gardoni scale) to describe the impact of different types of natural events and to facilitate the comparison.**

**Response:** Thank you very much for your summary. However, the objective of the paper is not to describe the impact of events. Instead, we propose the Gardoni Scale to directly compare the hazard strengths of events across hazard types. The hazard strength of an event is not the same as the impact of the event, as the impact is associated with not only the hazard strength but also the values exposed to the hazard strength and the vulnerability of the exposed entity to impact. By using a large sample size of data with a good quality, we can assume that the factors of exposed value and vulnerability have been controlled for such that we can use the observed impact to calibrate a regression model to quantify the hazard strength as the expected impact given average exposed value and vulnerability. Once such regression models are established for each hazard type, they can be used to derive the equivalent hazard strengths for direct comparisons across hazard types.

To clarify that our objective is not to describe the impact of events, we have modified the first paragraph of our introduction section. The modified first paragraph of the introduction section now reads: "Natural hazards pose significant challenges to human societies around the world. Between 2000 and 2020, natural hazard events caused over 130 billion dollars in losses and 64 695 fatalities, and affected more than 196 million people, on average each year (Guha-Sapir et al., 2021). Hazardous events, such as earthquakes, floods, and forest fires, can inflict heavy losses to communities when people and property are exposed to the natural forces of these events. The impacts of events, whatever their type, can be quantified directly (e.g., by financial loss; Hillier et al., 2015), or estimated on a scale. To estimate the impacts of an event with the consideration of its hazard strength, various impact scales have been proposed, including the Bradford disaster scale (Keller et al., 1992; 1997), unified localizable crisis scale (Rohn and Blackmore, 2009; 2015),

disaster impact index (Gardoni and Murphy, 2010), and cascading disaster magnitude (Alexander, 2018). However, a hazard strength scale is not the same as a hazard impact scale, as impacts are also driven by the exposure and vulnerability of entities, such as individuals, communities, and infrastructures, to an event. This makes it difficult to use impact scales to compare hazard strengths across natural hazard types. For example, the 2011 Christchurch earthquake was one of the most destructive earthquakes in New Zealand, albeit with a medium hazard strength of 6.2 in terms of its moment magnitude (Kaiser et al., 2012). Meanwhile, the 1964 Alaskan earthquake, with a larger moment magnitude of 9.2, resulted in fewer casualties and less economic damage than the Christchurch earthquake (United States Geological Survey [USGS], 2021)." (L26-39)

**Comment:** Although I do agree with the main idea of the paper, i.e., hazards cannot be compared but we can compare their effects, I have several doubts about this paper.**

Response: Thank you very much for your encouragement and comment. The comparison of hazards often involves the computation of the expected frequency of or the expected exceedance frequency of hazard strength for a given hazard type. In this paper, we aim to derive an estimate of hazard strength independent of hazard type. The main idea of the paper is that hazard strengths can be compared. First, hazard strengths can be compared within each hazard type. Agentially, for example, an M7 earthquake on the moment magnitude scale is larger than an M5 earthquake in terms of hazard strength, while the effects of the M5 earthquake can be much more severe than the effects of the M7 earthquake. Locationally, as another example, a community surrounded by a water depth of 2 meters is experiencing a much larger hazard strength of flood than another community facing a water depth of 0.5 meters. However, due to different pre-event mitigation efforts, the community with a 2-meter water depth may be impacted much less than the community with a 0.5-meter water depth. On the other hand, comparison of hazard strengths across different hazard types is difficult with the mainstream methodologies used in the existing multi-hazard studies. In light of this, the main academic contribution of this paper is that we propose a scale that enables cross-hazard comparison of hazard strengths in an agential and durational manner. We have also demonstrated how to compare the agential-durational hazard strengths across hazard types in this paper.

**Comment:** *I* will describe below only the most important ones (omitting other minor points), with the hope that they can be of some usefulness for the authors.

**Response:** Thank you very much for your encouraging comment.

**Comment:** 1. As just said, hazards cannot be compared but we can compare their effects; this is exactly what the risk analysis is meant to do. There is an extensive scientific literature on the comparison of the risks caused by different events (e.g., comparing the individual risk of death caused by different events), or comparing the risk with the acceptable risk that has been defined by decision makers. It is not clear why the authors dismiss completely all these efforts, which have eventually their same goal. Why do they think that their method is more effective that the classical risk and multirisk assessment?

**Response:** Thank you very much for your comment and question. As mentioned previously, the focus of the paper is on quantification of the equivalent hazard strength. As such, the objective is

not to perform a risk analysis. Instead, we are deriving a new indicator: equivalent hazard strength, that is derived from impacts given average exposure and vulnerability based on a robust record of historical impacts. While we agree that it is important to understand how different factors contribute to overall risk, this is not the goal of this paper. Our proposed scale has applicational significance in providing benchmark measures of hazard strength for vulnerability and resilience analyses. In addition, the derived equivalency of hazard strengths can be used to create multi-hazard hazard maps to show the distribution of exceedance probability of hazard strength across different hazard types.

To highlight that one of the main utilities of hazard equivalency research is to facilitate risk analysis, we have modified one sentence in the second paragraph of the Contributions section. The modified sentence now reads: "Such multi-hazard quantification of hazard, exposure, vulnerability, and resilience can be integrated to facilitate risk analysis to predict future losses and loss ratios without additional efforts to develop sophisticated models for each individual hazard types." (L348-350)

**Comment:** 2. The authors based their analysis on a 120-year-long database. I think that the length of this database is clearly too short to get a realistic estimation of the impact of some natural threats, which have a longer average inter-event times (for instance super-eruptions with VEI7 or 8). That's important because the effect of one of such events can largely overcome the cumulate effects of all other events. As a matter of fact, for some of the hazard considered in this paper, the most impacting events at worldwide scale have a return time that is much higher than 100 years. This is also the reason for what the risk is almost never empirically calculated using databases of this time length, at least for the most damaging events.

**Response:** Thank you very much for your comment. As mentioned previously, the objective of the paper is not to quantify the effects of events but to demonstrate the computation of the equivalency of hazard strengths. Having said this, we do agree that the length of the database used for our study may not be long enough for comparison across hazard types that occur infrequently. However, as the purpose of the paper is to propose an empirical method to derive the equivalent hazard strength across hazard types, the EM-DAT database is sufficiently robust for demonstration of the proposed method. When other databases with higher quality and longer length become available, we intend to use them to improve our model results. In terms of the issue associated with the return period, the return period is always positively correlated with the hazard strength independent of the hazard type. Analysis of return periods for small to medium hazard strengths provides evidence to extrapolate the return period for large hazard strengths that are rarely experienced. When dealing with large hazard strengths, it is common even for singular hazard strength scales to have trouble in revealing the hazard strengths. For example, all the earthquake magnitudes are known to have saturation issues, more or less to some degree, for large magnitudes. Therefore, it is expected that the derivation of equivalency of hazard strength for large hazard strengths may be less reliable than for smaller hazard strengths. Since most of the hazard damages communities experience are caused by events with small to medium hazard strengths, we believe that it is useful and significant to derive hazard equivalency even for small to medium hazard strengths. In addition, to what extent the computation of equivalent hazard strength becomes unreliable given a certain return period is beyond the scope of the current study. Future work should explore the effect of long return period on the estimation of hazard equivalency.

**Comment:** 3. I think that the exposure and vulnerability are strongly changing through time. Conversely, the authors are assuming that these quantities remain constant in the past 120 years. This assumption may introduce a significant bias in the ranking of the events; for instance, it may be argued that the same tsunami in 2004 would have caused much less casualties if it happened in 1904 (by the way, to my knowledge the number of casualties caused by the 2004 tsunami is much less than 2 millions as reported by the authors). Not less important, as also acknowledged by the authors, some of the data may be severely incomplete; incompleteness has to be carefully checked because it can introduce an important additional source of bias in the analysis.

**Response:** Thank you very much for your comment. We agree that exposure and vulnerability are changing through time. However, if such changes occur consistently across different hazard types, these changes are unlikely to affect the computation of equivalency of hazard strengths. By using the data of 120 years throughout the entire world, we assume that each hazard type has a similar temporal distribution of exposure and vulnerability, as is common in multi-hazard disaster research. Future work could examine whether this assumption holds true. Meanwhile, we also agree that there are some issues in the data from the EM-DAT database. In particular, there are many hazard types, such as volcanic hazards, that do not have measures of hazard strengths in the database. In addition, as also highlighted by the referee, there may be inaccurate records of disaster damages. In this study, we performed a quality control to exclude some obviously impossible values. We also excluded data points without hazard strength indicator values and performed a principal component analysis to support the derivation of the impact metric for data points with missing values for some, but not all, of the impact variables. However, we would like to point out that EM-DAT is a world-renowned database for hazard events and is one of the few open-source databases readily available to researchers and is often used for hazard analyses.

Regarding the reported 2 million casualties, it was incorrect, as it should be 2 million people affected by the tsunami. We have corrected this issue in the revised version of our manuscript. The modified sentence now reads: "The well-known 2004 Indian Ocean tsunami that affected more than 2 million people ranks 10th among all events, with its equivalent magnitude at 8.27." (L236-237). In addition, we have thoroughly double-checked the entire manuscript for such similar typos and have corrected another record in the text on the economic damage of the 2013 El Reno tornado that was based on information from Wikipedia and inconsistent with the data from EM-DAT database. The modified corresponding sentences now read: "Among the considered 12 hazard types, the natural hazard with the lowest maximum equivalent magnitude is tornado. The tornado event with the largest equivalent hazard magnitude (3.62) is the 2013 El Reno tornado in Oklahoma, USA. This tornado event led to a total damage of over 2 billion 2019 USD (Guha-Sapir et al., 2021)." (L240-242)

**Comment:** 4. I am puzzled by the inclusion of synthetic data to fill the "missing" data. This may be very dangerous, because the 'new' data have been generated assuming that the model used to generate them is correct. To sum, I do not understand the need to generate synthetic data and not using only the ones available. (but I may be missing something here)

**Response:** Thank you very much for your very insightful comment. We agree that using synthetic data always requires extreme caution. Therefore, we only included data points without missing values of impact variables and data points with missing values of one or two, but not all three, impact variables. There are no synthetic data points involved in the study. The purpose of filling

the missing values of the partially incomplete data points is not to generate synthetic data points, but rather to form a mathematical mapping between the one or two impact variables with recorded values to the impact metric that is the principal component of the three impact variables derived with the complete data points.

**Comment:** 5. The results of the correlation between impact metric and hazard (Figures 3 and 4) are largely not statistically significant (maybe except in a very few cases case, but we need also to take into account that the statistical significance has to take into account also the multiple tests). It is difficult for me to understand how we could use these relationship to rank the hazards in a meaningful way.

**Response:** Thank you very much for your comment. We agree that many of the estimates of model parameters are not statistically significant after the calibration of the regression models. However, the purpose of the regression modeling is not to provide statistical inference between hazard magnitude indicators and impact metric, nor is it to make predictions of impact metric with hazard magnitude indicators. Instead, the purpose is to provide a computational tool to map the hazard magnitude indicators to the equivalent hazard magnitude, which is correlated with the expected value of impact metric. In this sense, the lack of statistical significance or the wide spread of data points is not a problem for the computational methodology for deriving the equivalency of hazard strengths. In the paper, we present the statistics of model parameters mainly for reproduction of research results.

**Comment:** Figure 4 shows that, on average, the smaller the event the lesser the impact. This is already very well known but the large scatter of the logarithmic quantities implies that, for example, a large earthquake can cause no victims whereas a smaller one can cause a huge number of casualties. It depends on where the earthquake occur. For example, on average about 20 earthquakes of magnitude 7 or above occur worldwide per year, but only a very few of them in the last century caused more than 100,000 casualties, whereas most of them do not produce any casualty, or very few; the scatter in terms of casualties spans about 5 orders of magnitude for such a kind of events. This is a consequence of using an 'agential' approach, whereas the risk is intrinsically 'locational' (de facto, the exposure is strongly spatially clustered over the earth).

**Response:** Thank you very much for your careful observation and insightful comment. We do agree that events with large hazard strengths may result in small impact, whereas events with small hazard strengths may lead to large impact. However, this is not necessarily a consequence of using an agential approach. When looking at hazard strengths locationally, for example the modified Mercalli intensity, we can still find many cases where large hazard strengths associated with trivial or no damage due to low exposed value or high resilience of communities experiencing the large hazard strengths. Nevertheless, it is important to continue research in hazard equivalency both agentially and locationally.

**Comment:** 6. The example reported in the discussion highlights the problem with this method. The authors say that the cold wave in Oklahoma city in 2021 has the same hazard magnitude in the Gardoni scale as an earthquake of magnitude 7.5. I do believe that a magnitude 7.5 in Oklahoma city would have caused an impact that is several orders of magnitude larger than the

**impact caused by the cold wave; and no impact (or very limited) if the same earthquake occurred in a remote area.**

**Response:** Thank you very much for your comment. As mentioned previously, the objective of the paper is to quantify the equivalent hazard strength in an agential and durational way and not to compare the actual impact locationally. The cold wave event mentioned in the paper not only affected Oklahoma City but also many other parts of the Southern United States. The impact of the cold wave event within the entire spatial range of the event is recorded to be equivalent to the impact of Hurricane Harvey and the expected impact of a magnitude 7.5 earthquake. Regarding earthquake, it is very likely that a magnitude 7.5 earthquake in Oklahoma City would only affect a relatively small area around Oklahoma City. That is why, agentially, the 2021 cold wave event is equivalent in hazard magnitude to a magnitude 7.5 earthquake, but locationally, the hazard intensity of the 2021 cold wave event may not be equivalent to the hazard intensity of a magnitude 7.5 earthquake. The issue raised by the referee here is not a problem at all. Locationally, a magnitude 7.5 earthquake in Oklahoma City would have caused an impact that is several orders of magnitude larger than the impact caused by the cold wave in Oklahoma City. Agentially, however, the expected impact of a magnitude 7.5 earthquake, with its epicenter in Oklahoma City, within its spatial entirety would be equivalent to the expected impact of the 2021 cold wave event within its spatial entirety, i.e., the entire Southern United States.

---

## Author Response (AR2)

March 31, 2022

Editor
Natural Hazards and Earth System Sciences (NHESS)

Subject: Resubmission of revised article "Equivalent Hazard Magnitude Scale" with manuscript number nhess-2021-87.

Dear editor,

We thank you and the referees for careful review of our manuscript (nhess-2021-87) entitled "Equivalent Hazard Magnitude Scale". To address the comments from the review team, we have made a major revision to the manuscript.

A detailed account of how we addressed the comments from the referees is attached below this response letter in a point-by-point style. The major changes to the manuscript are summarized as follows:

1) We have modified the order of introduction to hazard types in the Methodology section.

2) We have significantly revised the paragraph on the rationale for using logarithmic transformation on some hazard magnitude indicators.

3) We have added material in the Discussion section regarding the limitation of the tsunami magnitude indicator currently adopted in the study.

The revised manuscript is now 9 645 words long and contains seventy-one references, eight figures, two tables in the main text, two tables in the appendix, six supplementary data files, and one supplementary video.

We look forward to hearing back from you regarding our revision.

Sincerely,

Yi Victor Wang, Ph.D.
Postdoctoral Fellow
Institute for Earth, Computing, Human and Observing
Chapman University
ywang2@chapman.edu

Antonia Sebastian, Ph.D.
Assistant Professor
Department of Earth, Marine and Environmental Sciences
University of North Carolina at Chapel Hill
asebastian@unc.edu

**Anonymous Referee #2**

We thank you very much for your comments and questions. In the following, we copy your comments in *italics* and follow with our response. The major changes to the manuscript are summarized as follows:

1) We have modified the order of introduction to hazard types in the Methodology section.

2) We have significantly revised the paragraph on the rationale for using logarithmic transformation on some hazard magnitude indicators.

3) We have added material in the Discussion section regarding the limitation of the tsunami magnitude indicator currently adopted in the study.

**Comment:** *I have already reviewed this paper in its original version. I do appreciate the attempt made by the authors in trying to answer to my queries. However, I still have the same doubts.*

**Response:** Thank you very much for your time and consideration regarding our manuscript in its original version and the revised version after the first round of review. In the response below, we make another attempt to clear your same doubts.

**Comment:** *Here I try to summarize the most critical comments in a concise way to clarify my view; I hope that this may be of some usefulness for the authors.*

**Response:** Thank you very much for your summary. We appreciate and value your critical comments.

**Comment:** *Hazard analysis tries to forecast what the Nature is going to do; and the risk quantifies the impact of those events on humans and the environment.*

**Response:** Thank you very much for your comment. In the field of science, engineering, and management of hazards and disaster risks, the words "hazard" and "risk" are often used interchangeably to refer to different or the same conceptual domains. Etymologically, "hazard" and "risk" have few differences. According to Möller 2012 (https://doi.org//10.1007/978-94-007-1433-5_3), "risk" can be mainly used to refer to five conceptual domains, which we can further summarize into three: 1) the source/cause of unwanted event or situation (which is also called the hazard agent), 2) a risk quantity referring to the expected loss and/or associated probability distributions, and 3) the unwanted event or situation itself. If we look at literature and academic communications, it is not difficult to find the cases where the word "hazard" is used to describe the same three conceptual domains of "risk", i.e., risk source (hazard agent), risk quantity, and risk situation. Therefore, the terms "hazard analysis" and "risk analysis" can literally refer to exactly the same thing. Likewise, "hazard" and "risk" can literally quantify exactly the same thing. The differences in the use of these terms in practice are to a large extent subjectively, arbitrarily, and authoritatively (instead of being objectively, logically, and scientifically) determined and recognized by different groups of scholars.

**Comment:** *The hazards cannot be compared (how can we compare the ash fall loading of a volcanic eruption, ground shaking caused by an earthquake, or the run up of a tsunami?); the risk can be compared because they may refer to the same loss metric (number of human lives, economic losses, or else).*

**Response:** Thank you very much for your comment and question. Since "hazard" and "risk" are often used interchangeably in the public and academic domains, we need to be careful when we are using these words to describe quantitative measures. Here, we use the term "hazard strength" to refer to the measure of the force of a hazard agent associated with an event potentially resulting in adverse impacts. We agree that the impact and the expected impact (or "risk") can be compared across hazard types, as they may refer to the same loss metric. However, we disagree with the statement that the forces of hazard agents cannot be compared (e.g., ash fall loading of a volcanic eruption versus ground shaking of an earthquake). In fact, in this manuscript, we have demonstrated that it is possible to compare the strength of a hazard associated with events across different hazard types by using the expected impact given average exposed entities. It is worth noting that the expected impact given average exposed entities is not the same thing as merely the impact. We are not using the recorded impact to directly compare forces of hazard agents.

**Comment:** *It is not clear to me what is the "hazard strength". The authors write "... equivalent hazard strength, that is derived from impacts given average exposure and vulnerability based on a robust record of historical impacts.". So, it is not a hazard because it is calculated from the "average" impact of the event, but the authors claim that is neither the impact, even though it is linearly estimated by it.*

**Response:** Thank you very much for your comment. As mentioned previously, hazard strength is the force associated with the source of an event that may result in adverse impact prior to incorporating vulnerable entities exposed to the force. Hazard strength could therefore be a measure or function of ash fall loading of a volcanic eruption, ground shaking caused by an earthquake, or the run up of a tsunami, as the reviewer has suggested. Based on a systematic review of existing hazard strength metrics, such as earthquake moment magnitude, Saffir-Simpson hurricane wind scale, and tornado Fujita scale, included in another submission, we have noticed that these hazard strength metrics can be further categorized into four groups along two conceptual dimensions, i.e., the spatial dimension and the temporal dimension. Accordingly, the four groups include agential-durational metrics, location-durational metrics, agential-momental metrics, and locational-momental metrics. All hazard strengths can be categorized into one of these four hazard types. However, there are two major challenges to the comparison of hazard strengths across different hazard types.

First, it is only meaningful to compare the hazard strengths of the same spatial-temporal type. This can even be demonstrated by the different types of hazard strengths for a same hazard type. For example, the comparison between a moment magnitude and modified Mercalli intensity for earthquakes is meaningless, as the former is an agential-durational metric while the latter is a locational-durational one. However, it is possible and common to compare the Richter magnitude and the moment magnitude because both are agential-durational metrics.

Second, there is no existing method to quantify the equivalency of a hazard strength when comparing hazard strengths across hazard types. In this manuscript, we propose a methodology to compare the agential-durational hazard strengths across different hazard types by deriving the

equivalency of these hazard strengths. To do so, we can build quantitative models to map considered hazard strengths to a common scale, which we define as the equivalent hazard strength scale. Such models can be calibrated by targeting the expected impact given average vulnerable entities exposed to the hazard strength. Even though the target can be considered as the average impact as pointed out by the reviewer, it is not the actual impact. The expectation of a variable and the variable itself are quantitatively and qualitatively different. Thus, it is problematic to assume that the average/expectation/mean of the variable and the variable itself are interchangeable since they refer to different conceptual domains. Here, by using the expectation of adverse impacts, we have demonstrated that it is possible to derive the equivalency of hazard strengths of events across different hazard types.

**Comment:** 2. *So, if it is not clear what the hazard strength means, how can it be used in practice?*

**Response:** Thank you very much for your comment. For clarity of the meaning of hazard strength, please see our response to the previous comment. The idea and utility of hazard strength is not new. However, what is novel in this paper is the summarization regarding the types of hazard strengths and how to derive the equivalency of hazard strengths across different hazard types. Regarding the utility of the concept of hazard strength, we are seeing it everywhere. When news media are reporting on an M8 earthquake, that "M8" refers to the hazard strength (an agential-durational one) of the earthquake event. When news media are reporting on a hurricane evolving from Category 1 to Category 5, that "Category 1" and "Category 5" are the measures of hazard strength (an agential-momental one) of the hurricane. However, it is difficult for the general public, or even emergency managers to understand how to prepare for hazard events that have not yet been experienced. Particularly in a new and more violent world (e.g., due to climate change), the ability to compare hazard strengths across multiple hazard types has immense value (see, e.g., the tornado experienced in New Orleans recently).

**Comment:** *In other words, what is the additional information carried out by the hazard strength with respect to hazard and risk?*

**Response:** Thank you very much for your comment. As mentioned previously, "hazard" and "risk" are often used interchangeably. To avoid confusion and to facilitate scientific communication, we try to find and use a term to refer to specifically the force associated with the hazard agent of an event that could result in adverse impacts when vulnerable entities are exposed to such a force. There are candidates such as "magnitude" and "intensity". Based on our systematic literature review for another publication, however, we have noticed that these two words, "magnitude" and "intensity" are used to refer to different types of hazard strengths along the conceptual spatial and temporal dimensions across hazard types, even though for each individual hazard type, the use of "magnitude" and "intensity" may follow certain rules (e.g., in earthquake research). Therefore, for our purposes, we find it necessary to use another term to refer to things such as "magnitude" and "intensity" and we settle down with the term "hazard strength".

**Comment:** *The authors write "Our proposed scale has applicational significance in providing benchmark measures of hazard strength for vulnerability and resilience analyses". That's exactly*

*what the risk is meant to do. Why may the use of hazard strength provide additional relevant information on this point?*

**Response:** Thank you very much for your comment and question. First, "vulnerability" and "resilience" can be used to refer to multiple conceptual domains. Here, for demonstration as an example, let's use the word "vulnerability" to refer to the tendency to suffer adverse impacts given a certain level of hazard strength. Such a vulnerability can be manifested as the casualty ratio, building damage ratio, or economic loss ratio given a certain level of hazard strength. Suppose there are two communities. community A has just experienced an M5 earthquake while community B has just experienced a C5 hurricane. Suppose both communities A and B lost 10% of their population due to the earthquake and the hurricane, respectively. Can we say that these communities seem to have the same vulnerability to adverse impacts because they experienced the same loss ratio? Probably not because we still need to know how to compare the "M5" and "C5". Can we use the existing risk modeling efforts to compare "M5" and "C5"? Probably not because they are not capable of doing so. In this case, then, we can use our proposed concept of equivalent hazard strength, which is what we mean by the benchmark measures to support vulnerability and resilience analyses.

**Comment:** *In my understanding, the authors claim that the hazard strength is a spatially averaged quantity whereas the risk is more localized. This is not true, because we can calculate the risk at any spatial scale, for example we can calculate the probability that one specific loss (e.g., 1 billion of euros or more) will be caused in the next 10 years by earthquakes, tsunamis, floods, or else, in Europe or at any spatial region.*

**Response:** Thank you very much for your comment. However, the reviewer's understanding is incomplete. Hazard strength is not necessarily a spatially averaged quantity. As mentioned previously, there are four types of hazard strength, i.e., the agential-durational metric, the locational-durational metric, the agential-momental metric, and the locational-momental metric. Only the agential-durational and agential-momental metrics are spatially aggregated quantities. The other two types are specific to locations. In this manuscript, we only present the agential-durational type of equivalent hazard strength, which is a spatially aggregated metric. Our typology on the 4 types of hazard strength can actually also be applied to impact metrics and risk metrics.

**Comment:** *Again, not very clear what the hazard strength tells us more than what we already know.*

**Response:** Thank you very much for your comment. The contribution of our study is on the equivalent hazard strength, not merely on the hazard strength. In our study, we have demonstrated that it is possible to compare hazard strengths of events across different hazard types. We have also proposed a quantitative modeling framework for derivation of equivalency of hazard strength across hazard types.

**Comment:** *The authors also claim that "In addition, the derived equivalency of hazard strengths can be used to create multi-hazard hazard maps to show the distribution of exceedance probability of hazard strength across different hazard types." That's a bold statement, but to me it is not clear at all how this measure can be used for multi-hazard purposes (cascading events, etc…)*

**Response:** Thank you very much for your comment. First, before we reach cascading events, it's important to consider the aggregation of expected frequency distributions of hazard strengths of different hazard types for hazard mapping. Existing practices regarding such an aggregation are simply summation without any justification for the weights used for each hazard type. With the derivation of equivalent hazard strength, we can convert different hazard strengths onto a same scale and aggregate the influences of these hazard strengths by using the equivalency of hazard strengths to create a multi-hazard hazard map involving multiple types of hazards. This will be a significant improvement to the current modeling effort for a multi-hazard hazard map. Then, regarding cascading/compound events, our next step would be to apply the typology to a minimum of four types of hazard strength to systematically develop methods to define the hazard strength metrics and to measure these hazard strengths of cascading/compound events before we can develop and train models to compute the equivalency of hazard strengths for these cascading/compound events in a way that is consistent with what we have proposed. Once this is achieved, cascading/compound events can also be included in a multi-hazard hazard map.

**Comment:** *The authors agree that the database used is not long enough to have a complete view of the losses that can be caused by all natural events (most of them having a long return period), but they claim that "... as the purpose of the paper is to propose an empirical method to derive the equivalent hazard strength across hazard types, the EM-DAT data base is sufficiently robust for demonstration of the proposed method."*

**Response:** Thank you very much for your comment. This manuscript presents a pioneering study to propose the concept of equivalency of hazard strength and how to compute it. Therefore, the EM-DAT database is sufficient for our purposes herein. However, to systematically compute the equivalent hazard strengths across all four different types along the conceptual spatial and temporal dimensions and across major hazard types including cascading/compound events, we need a database with much higher quality in terms of records of indicators of hazard strengths of events than any existing databases of disaster events. Such an ideal database will also need to include events with zero or almost zero losses and cover a long period of time to account for events with a long return period as much as possible.

**Comment:** *Moreover, they calculate the hazard strength using a regression analysis on parameters that are not statistically correlated. The authors write "However, the purpose of the regression modeling is not to provide statistical inference between hazard magnitude indicators and impact metric, nor is it to make predictions of impact metric with hazard magnitude indicators. Instead, the purpose is to provide a computational tool to map the hazard magnitude indicators to the equivalent hazard magnitude, which is correlated with the expected value of impact metric.". In my opinion, this view is quite debatable. What is the scientific reliability of a measure that is obtained by a non statistically significant regression? And using a too short catalog?*

**Response:** Thank you very much for your comment and questions. Among our regression models for equivalent hazard magnitude, many are of statistical significance, e.g., for earthquake and tropical cyclone. Some are not due to a lack of data points, e.g., for cold wave and heat wave. However, the purpose of the study is not to make statistical inference nor predictions of adverse impacts, but rather to provide a quantitative tool to convert hazard strengths to an equivalent hazard strength across different hazard types. The scientific significance of a study such as this presented

one is relative to its purpose. Indeed, we agree that, to improve the scientific reliability, we need better data to support our modeling so that we may have a sufficiently large amount of data points to obtain statistically significant results for all considered hazard types and we have proposed this as future work.

Moreover, our study demonstrates the need for database of disaster events with a much higher quality in terms of indicators of hazard strength metrics and impact metrics, which would result in more data points for improvement of the modeling in terms of scientific reliability not only for equivalency of hazard strength but also further for empirical disaster vulnerability and many other metrics that can be used for disaster risk reduction.

Regarding the short catalog issue. Then the question also goes to "how short is too short?" and "how long is long enough?" With a database with data of 1,000 years, we would still miss data points for events once in 1,000 or 2,000 years. The tail distribution of the frequency of extremely large events is always a challenge no matter how one models it. Regardless, with a database with data of hundreds or even only tens of years, we can already develop decent models for those events with shorter return periods, which actually correspond to most of the disaster losses that communities are likely to experience. Nevertheless, when high-quality data of a long period such as thousands of years become available, we would like to use them as well.

**Comment:** *Note that the length of the catalog is very important, because the inclusion of just one single future event (e.g. a very large volcanic eruption) may severely impact the hazard strength.*

**Response:** Thank you very much for your comment. This manuscript presents a pioneering study in equivalency of hazard strength. It opens the door for future opportunities in this new area of research. We agree that the inclusion of outlier events may affect the computation of equivalent hazard strength especially for hazard type with a limited amount of data points included in a database. Future work should systematically study the effect of outliers and attempt to develop more robust computational methods for potential improvement.

**Comment:** *My last comment on the original manuscript, where I discuss the comparison of the 2021 cold wave in US with an earthquake of magnitude 7.5 is not satisfactory (maybe because I was not clear enough in my previous comment). I understand the spatial dimension of the events is very different. But the point is that these two events may have very large different impacts. For example, if the same cold wave would have impacted the east coast instead of Texas and Oklahoma. Or if a M75 would occur inside a large city in US or in the middle of a desert. To sum, I still have may doubts on the usefulness of a scale that equates a cold wave (as the 2021 event) with a M7.5 earthquake in terms of "strength haard", simply because in a short database these two events seem to have produced on average the same losses.*

**Response:** Thank you very much for your comment. We agree that different events may result in different impacts. However, this can also manifest for events of the same hazard type. For example, suppose that earthquake A is only recorded with a Richter scale at $M_L 7.5$. Meanwhile, earthquake B is only recorded with a moment magnitude scale at $M_W 7.5$. We know earthquake A and earthquake B are equivalent to each other in terms of the agential-durational hazard strength because one is $M_L 7.5$ and the other is $M_W 7.5$, but their impacts can be totally different. Here, the equivalency of hazard strength refers to the expectation of impact given average vulnerable entities

exposed to the hazard strength, not the observed impact. We welcome the concerns and conversation from the reviewer, but our proposed equivalency of hazard strength is derived from the expected/average losses of all events of a same hazard type.

**Comment:** *I have many other doubts, but in general I think that the approach adopted by the authors is vastly speculative and its usefulness is not clear, in particular it is not clear what is the advantage of the additional information provided by the hazard strength with respect to the hazard and risk analysis.*

**Response:** Thank you very much for your comment. In general, we believe our study provides a solid empirical methodology to compare hazard strengths of events across different hazard types. Such a methodology along with its corresponding typology on hazard strength advances existing hazard and risk analysis. Therefore, we believe it has merit and deserves publication.

**Anonymous Referee #3**

We thank you very much for your constructive comments and insightful suggestions. In the following, we copy your comments in *italics* and follow with our response. The major changes to the manuscript are summarized as follows:

1) We have modified the order of introduction to hazard types in the Methodology section.

2) We have significantly revised the paragraph on the rationale for using logarithmic transformation on some hazard magnitude indicators.

3) We have added material in the Discussion section regarding the limitation of the tsunami magnitude indicator currently adopted in the study.

**Comment:** *The manuscript reports an interesting attempt to define a common indicator of the magnitude of the agents responsible for different hazard types, in order to make different hazards comparable in the framework of a multi-risk management. The definition of a common metrics through very different phenomena presents several difficulties, thus this study can be considered a first attempt to define a methodology for comparing the severity of different hazards. However, in my opinion, some further efforts can be done to reduce the uncertainty factors affecting the definition of a common hazard estimator. In the following I report some observations and suggestions for possible improvement and for the discussion of the uncertainty factors, together to a few minor comments.*

**Response:** Thank you very much for your summary and encouragement. We have made revisions according to your comments and suggestions as follows.

**Comment:** *Lines 121-126: The authors report that, as hazard magnitude indicator for earthquake and tsunami, they used Richter magnitude (also known in literature as Local Magnitude - ML), in that this is the parameter provided by the catalogue employed (EM-DAT). In section 5.2 they discuss some problems with regard to the use of hazard indicator for earthquakes and tsunami, recalling that Richter magnitude is an unsatisfactory parameter. Actually, a more appropriate indicator would be the moment magnitude (Mw), since Richter magnitude tends to saturate for magnitudes above 6.5. However, after having examined the EM-DAT catalogue, I suspect that the magnitude values in this catalogue are improperly defined as "Richter". For instance, for the 1977 Sumba earthquake (Indonesia) the reported value (8) actually is the moment magnitude rounded to an integer values (the USGS estimate of Mw for this earthquake is 8.3), which for sure is not an ML value.*

**Response:** Thank you very much for your comment. We acknowledge that Richter magnitude is also usually called the local magnitude $M_L$. We also acknowledge that the Richter magnitudes and moment magnitudes ($M_W$) may be mixed in the EM-DAT database. However, for any earthquake event, its $M_L$ and $M_W$ is usually similar. The EM-DAT database rounds the earthquake magnitudes to integers. Once rounded to integers, most of the earthquakes have the same $M_L$ and $M_W$. While there may be mistakes in the EM-DAT records such that some of the Richter magnitudes are actually moment magnitudes, identifying and correcting these errors is beyond the scope of this study. Here, we follow and keep a consistent protocol for data processing across different hazard

types, i.e., following what EM-DAT has recorded for all considered hazard types, in order to demonstrate the application of our proposed method for identifying equivalent hazard magnitude.

**Comment:** *There is, however, an additional problem to consider: the magnitude values reported in the EM-DATA catalogue appears all rounded to integer values, which represent a rough approximation (an estimate error of 0.5 correspond to an uncertainty of energy by a factor ~5). An improvement of data quality could be easily obtained by replacing such integer magnitudes with the estimates provided for the same events by international catalogues (for instance the USGS significant earthquake catalogue -*
*https://earthquake.usgs.gov/earthquakes/browse/significant.php).*

**Response:** Thank you very much for your comment. We agree that the rounding of the earthquake magnitudes could be problematic for quantitative modeling. However, because the purpose of the study is to demonstrate an approach to identify equivalent hazard magnitude, rather than predicting the adverse impact of an event, we feel that our use of the rounded earthquake magnitudes is sufficient for this analysis. However, we also recognized that the rounding of magnitude records may introduce some uncertainty to the data used in our analysis. To address this concern, we would highlight that the computation of equivalent hazard magnitude is based on the expectation line of the impact metric and such an expectation line (or the regression line) manifests the central tendency (or the first moment) of the data points and is minimally affected by the increase of uncertainty in terms of the variation (or the second moment) of data points.

In response to your comment regarding other hazard catalogues, we recognize that considering other catalogues may improve the quality of data points. However, because we wanted to follow a consistent protocol for all considered hazard types to demonstrate the application of the methods, we chose to only include events and the data points within the EM-DAT catalogue. Assuming that our publication is successful, we plan to conduct additional studies to systematically improve the proposed equivalent hazard strength methods. We also welcome others to propose such studies. However, prior to doing so, it will be necessary to improve the quality of data on hazard/risk/disaster events, especially in terms of the records of indicators of hazard strengths and the inclusion of events of zero or almost zero losses, as you have suggested.

With respect to earthquake event databases, the first author of this study, YVW, has previously examined several earthquake event catalogues/databases and used their records for his empirical modeling of disaster vulnerability (see, e.g., Wang et al. 2019 https://doi.org/10.1193/022618EQS046M, 2020 https://doi.org/10.1061/(ASCE)NH.1527-6996.0000356, 2021 https://doi.org/10.1080/24694452.2020.1823807). These catalogues include the USGS one recommended by the reviewer. During these previous research works, he has noticed that the records of earthquake events do not always match across different databases. There are records in one database that do not correspond to the records in another database.

**Comment:** *With regard to tsunamis, one should consider that these phenomena can be also caused by non-seismic sources (e.g. volcanic island collapses and large coastal landslides) and that, in case of earthquake-induced tsunamis, the energy transmitted to sea waves does not depend only on the magnitude of the triggering earthquake but also on its focal mechanism (strike-slip mechanisms generally are not tsunamogenic) and depth of the fault rupture. Thus the magnitude of the earthquake generating tsunami is a rather rough indicator of tsunami energy and, actually,*

*some proposals exist to measure a specific "tsunami magnitude" (see Papadopoulos et al., 2020 - https://doi.org/10.1016/B978-0-12-8156686-5.00003-1 - for a review).*

**Response:** Thank you very much for your comment and suggestion. We acknowledge that the use of Richter magnitude as the hazard magnitude indicator for tsunami is debatable because tsunami can also be triggered by submarine volcanic activities and large coastal or submarine landslides. We also acknowledge that only underwater megathrust earthquakes can generate tsunami. However, for our purposes, whether an earthquake is a thrust-fault one or not does not affect the use of Richter magnitude for tsunami. To indicate the limitation of use of Richter magnitude for tsunami, we have added one sentence in the Discussion section after the limitations of earthquake hazard magnitude indicators.

The added sentence reads: "In addition, regarding tsunami, the mere inclusion of Richter magnitude of a tsunami-triggering earthquake as the magnitude indicator ignores the fact that tsunami can also be caused by non-seismic events, such as volcanic island collapses and large coastal landslides." (L374-376)

We would also like to point out that this manuscript is one of two recent submissions on the equivalent hazard strength scales for peer-review. This paper focuses on the agential-durational scale, while the other is on the locational-durational scale. In that paper, we undertook a systematic literature review of 69 existing hazard strength metrics for 21 singular hazard types and for multiple hazards. These metrics include the Abe magnitude (Abe 1979), the Murty–Loomis magnitude (Murty and Loomis 1980), the Sieberg–Ambraseys intensity (Ambraseys 1962), the Imamura–Iida intensity (Shuto 1993), and the most recently developed integrated tsunami intensity (Lekkas et al. 2013) for tsunami.

**Comment:** *Lines 132-137 – The authors declare to have applied logarithm transformation to the hazard magnitude indicators of eight hazard types so to have a value distribution close to Gaussian in the range ±∞. Such transformation was not applied to temperature-based (heat/cold waves) and magnitude (earthquake/tsunami) indicators. I believe that the appropriateness of applying or not this transformation should be evaluated by examining if the impact parameters (or their logarithms) show a dependence closer to linear if expressed as function of the logarithm of the hazard magnitude indicators instead of the untransformed original indicators. The proposed justification for not applying logarithmic transformation to strength indicator of earthquakes/tsunamis and heat/cold waves, i.e. the assumption that magnitudes and temperatures have already a range similar to ±∞, seems to me rather questionable. Indeed, while the logarithm of impact indicators can assume values ranging from –∞ (zero damage) to a maximum (which however is not infinite), the atmospheric temperature at the earth surface cannot assume values in the range (–273.15, ∞), but only, approximately, in the order of –60° - 0° for cold waves and 30° - 60° for heat waves. With regard to earthquake magnitude, damaging earthquakes start only from magnitude ~4.5 and magnitudes larger than ~10 are probably not possible because rock strength does not allow energy to be stored, before rupture, up to generate earthquakes stronger than such a magnitude (the largest recorded magnitude is 9.5). A more tenable justification for not applying a logarithmic transformation to earthquake magnitude is that it is already a logarithmic parameter (it is derived from the logarithm of the maximum ground motion amplitude or of the seismic moment). In general, a logarithmic transformation appears reasonable for parameter spanning through several orders of size, which, for instance, is the case of the seismic ground motion amplitudes but not of*

*earthquake magnitudes. I believe that the application of logarithmic transformation to hazard magnitude indicators should be more thoroughly discussed and more soundly motivated.*

**Response:** Thank you very much for your comment and suggestions. The logarithmic transformation to a hazard magnitude indicator is applied mainly to make the distribution of transformed data points to be consistent with a linear model such that a linear regression line can be used as the expectation of impact metric for construction of the equivalent hazard magnitude for the corresponding hazard type. Meanwhile, the logarithmic transformation is also to ensure that any new input hazard magnitude indicator value can correspond to an equivalent hazard magnitude value. For this purpose, the range of the transformed indicator value also needs to be spanned as much as possible so that the zero magnitude points for different hazard types are close to each other at $-\infty$ regarding the transformed indicator and there also remains a possibility of infinite magnitude corresponding to the point of theoretically infinite loss. Arbitrarily setting a lower bound and an upper bound for such a range is problematic. For example, the range of [–60, 0] for cold wave and the range of [30, 60] for heat wave should not replace the possible range of $(-273.15, \infty)$ for temperature. Here, when we need to consider a new data point at –68° for cold wave or at 25° for a heat wave, we should still be able to provide an estimated value of the equivalent hazard magnitude for such an event (although the meanings of such estimates may deserve discussions in future work). For earthquake, as another example, the range of magnitude should not be [4.5, 10]. Even though the largest recorded earthquake magnitude is around 9.5 on earth, there are possibilities that M10 or even M11 earthquakes could occur in the future under extremely rare planetary impacts. Considering these, however, it is also worth mentioning that the logarithmic transformation is not for stabilizing variances in the data points. Because the objective of this study is to use the expected impact metric given average vulnerable entities exposed to hazard magnitude to derive the equivalent hazard magnitude, the variance of a hazard magnitude indicator of data points is not affecting the computation of equivalency of hazard strengths at all. To clarify the main purpose of application of logarithmic transformation to some hazard magnitude indicators, also taking into consideration the suggestions from the reviewer, we have modified the last paragraph of the Data Collection subsection.

The modified Data Collection subsection now reads: "To facilitate regression modelling, we logarithmically transformed values of hazard magnitude indicators to be close to a Gaussian distribution within the theoretical range $(-\infty, \infty)$ for eight of the hazard types. Such logarithmic transformations were conducted to keep the shape of distribution of data points consistent with their corresponding linear regression models. The indicators that were not logarithmically transformed included minimum temperature of cold waves, maximum temperature of heat waves, Richter magnitude of earthquakes, and earthquake Richter magnitude of tsunami. Cold wave and heat wave events were excluded from logarithmic transformations because the distributions of data points of these events did not present non-linear patterns and the Celsius temperature has a range $[-273.15, \infty)$ similar to $(-\infty, \infty)$. Meanwhile, the earthquake Richter magnitude is already a logarithmic metric with the desired theoretical range of $(-\infty, \infty)$." (L129-136)

**Comment:** *Lines 64-65: Proponing to call the new scale as "Gardoni scale", the authors should briefly explain the reason of this choice (e.g. the contribution of Gardoni to the advancement of this kind of studies).*

**Response:** Thank you very much for your comment. As explained in our response to Reviewer 1 during the first round of open review, the proposed agential-durational equivalent hazard strength scale is named in honor of Prof. Paolo Gardoni at the University of Illinois at Urbana–Champaign. This paper with NHESS is one of two manuscripts recently submitted for peer-reviewed journal publication on equivalent hazard strength. In the other submission, we include a systematic review of 69 existing hazard strength metrics for 21 singular hazard types as well as for multiple hazards. In the other paper, we also introduce in detail the typology on four types of hazard strength metrics, i.e., the agential-durational metric, the locational-durational metric, the agential-momental metric, and the locational-momental metric. To differentiate between the agential-durational and the locational-durational metrics more easily, the first author YVW would like to name these two metrics after his two doctoral co-advisors, Prof. Paolo Gardoni (PG) and Prof. Colleen Murphy (CM), considering that the original idea of equivalency of hazard strengths emerged during one of YVW's doctoral advisory meetings in 2015 with PG and CM in PG's office at the University of Illinois at Urbana–Champaign. This manuscript for NHESS is on the agential-durational equivalent hazard strength metric, which is suggested to be named after PG. The other manuscript is on the locational-durational equivalent hazard strength metric named after CM and has been accepted for publication. In this regard, it would seem to be appropriate to name the agential-durational hazard strength metric after PG. To clarify the reason for naming the agential-durational hazard strength metric after PG, we have slightly modified the corresponding sentence in the Introduction section to emphasize the phrase of "in honour of".

The modified sentence now reads: "The proposed scale is named in honour of the Alfredo H. Ang Family Professor Paolo Gardoni at the University of Illinois at Urbana–Champaign." (L63-64)

**Comment:** *Line 121-125: Introducing the list of discussed hazards, I suggest to re-arrange them, grouping hazards having analogies and similar magnitude indicators, e.g.: convective storm, extra-tropical storm, tornado, tropical cyclone (which use wind speed as hazard magnitude parameters), cold wave, heat wave (using temperature), drought, forest fire, flash flood, riverine flood (using affected area), earthquake and tsunami (using magnitude).*

**Response:** Thank you very much for your very insightful comment. We agree with your suggestion and have made corresponding modifications to the second paragraph of the Data Collection subsection.

The modified second paragraph of the Data Collection section now reads: "The 12 considered hazard types include convective storm, extra-tropical storm, tornado, tropical cyclone (wind speed is used as hazard magnitude indicator), cold wave, heat wave (temperature), drought, flash flood, forest fire, riverine flood (affected area), earthquake, and tsunami (Richter magnitude). For data quality control, we removed data points with questionable values of hazard magnitude indicators. For cold wave events, we only included data points with a minimum temperature $\leq 0$ °C; for convective storms, we only considered data points with a peak gust wind speed $\geq 60$ km h$^{-1}$; for forest fires, we only included data points with a burnt area $\leq 200$ thousand km$^2$; for heat wave events, we only considered data points with a maximum temperature $\geq 35$ °C and $\leq 57$ °C; for tornadoes, we only included data points with a peak gust wind speed $\geq 100$ km h$^{-1}$; and for tsunami, we only considered data points with an earthquake Richter magnitude $\geq 6$." (L121-128)

**Comment:** *5. From the diagram of Figure 5, an earthquake of magnitude 3 appears to have caused significant damages. This data is hardly credible, unless some secondary effect (e.g. a landslide triggered by the earthquake) was responsible of damages (but, in such a case, damages cannot be attributed to the earthquake magnitude). I suggest to remove this data from the analysis.*

**Response:** Thank you very much for your comment and suggestion. We have double checked this data point of M3. It corresponds to the record of an M3.2 event (https://earthquake.usgs.gov/earthquakes/eventpage/usp000923y/executive) on February 1, 1999, in Southern Russia from the USGS earthquake catalogue. This earthquake was subsequent to two other recorded nearby earthquakes on the same day, i.e., an M4.2 (https://earthquake.usgs.gov/earthquakes/eventpage/usp000921w/executive) and an M4.5 (https://earthquake.usgs.gov/earthquakes/eventpage/usp0009233/executive). In the EM-DAT database, there is only one record on this earthquake series, which corresponds to a rounded magnitude value of M3. Based on our data protocol, we strictly follow the EM-DAT record instead of using the M4.2 or M4.5 record for the magnitude indicator of this event. In addition, although the traditional belief is that damaging earthquakes need to be at least M4.5, small earthquakes do sometimes result in somewhat significant losses to vulnerable communities based on YVW's previous examinations of earthquake data (e.g., Wang et al. 2019 https://doi.org/10.1193/022618EQS046M, 2020 https://doi.org/10.1061/(ASCE)NH.1527-6996.0000356, 2021 https://doi.org/10.1080/24694452.2020.1823807). Also, as listed in https://www.pennlive.com/news/2019/06/how-strong-is-a-34-magnitude-earthquake-the-richter-scale-explained.html, M3-M3.9 earthquakes rarely causes any damage, and being rare is not being never. Rarely causing damage means that there is at least a possibility of damage. Beside the possibility of damage by an M3.2 earthquake, this data point is not far away from the regression line. Thus, it does not significantly affect the modeling result. Therefore, we keep this data point in our model, even though we deeply appreciate the reviewer's suggestion.

---

## Author Response (AR3)

October 29, 2022

Editor Natural Hazards and Earth System Sciences (NHESS)

Subject: Resubmission of revised article "Equivalent Hazard Magnitude Scale" with manuscript number nhess-2021-87.

Dear editor,

We thank you and the referees for careful review of our manuscript (nhess-2021-87) entitled "Equivalent Hazard Magnitude Scale". To address the comments from the review team, we have made minor revisions to the manuscript.

A detailed account of how we addressed the comments from the referees is attached below this response letter in a point-by-point style. The major changes to the manuscript are summarized as follows:

- 1) We have modified the Discussion section to address the referees' comments.
- 2) We have corrected the typos.
- 3) We have updated the references.

The revised manuscript is now 9 828 words long and contains seventy-one references, eight figures, two tables in the main text, two tables in the appendix, six supplementary data files, and one supplementary video.

We look forward to hearing back from you regarding our revisions.

Sincerely,

Yi Victor Wang, Ph.D. Postdoctoral Fellow Institute for Earth, Computing, Human and Observing Chapman University ywang2@chapman.edu

Antonia Sebastian, Ph.D. Assistant Professor Department of Earth, Marine and Environmental Sciences University of North Carolina at Chapel Hill asebastian@unc.edu

**Anonymous Referee #3**

We thank you very much for your comments and suggestions. In the following, we copy your comments in *italics* and follow with our response. The major changes to the manuscript are summarized as follows:

- 1) We have modified the Discussion section to address the referees' comments.
- 2) We have corrected the typos.
- 3) We have updated the references.

**Comment:** The revised version of the manuscript accepted some of my comments. However I have still some observations about the authors' replies to a couple of the issues I raised.

**Response:** Thank you very much for your time and consideration to help us improve the quality of this manuscript.

**Comment:** With regard to the possible misnaming in the EM-DAT catalogue of moment magnitude MW as "Richter magnitude" ML, the authors replied that "for any earthquake event, its ML and MW is usually similar" and "once rounded to integers, most of the earthquakes have the same ML and MW". In my opinion, this is not correct: some comparative studies pointed out possible underestimate by up to 2 units for ML in comparison to MW for earthquakes of magnitude 8 (cf. Chen & Chen 1989, https://doi.org/10.1016/0040-1951(89)90205-9), which, given the logarithmic nature of these parameters, is a rather large difference. The authors could resolve this question specifying that the EM-DAT catalogue reports generically as "Richter magnitude" estimates which likely include moment magnitude as well.

**Response:** Thank you very much for your suggestion. We have modified the Discussion section to emphasize in the revised manuscript that "the EM-DAT database reported generally as "Richter magnitude" estimates for earthquake events" (L371-372).

The corresponding sentences of the Discussion section now read: "Selection of hazard magnitude indicators in this study was also limited by the adopted datasets. As an example, the earthquake Richter magnitude (Richter, 1935) was the only recorded hazard magnitude indicator in the datasets of this study. However, the EM-DAT database reported generically as "Richter magnitude" estimates for earthquake events, even though such estimates may include moment magnitude as well. In addition, regarding tsunami, the mere inclusion of earthquake magnitude of a tsunami-triggering earthquake as the magnitude indicator ignores the fact that tsunami can also be caused by non-seismic events, such as volcanic island collapses and large coastal landslides." (L370-376)

**Comment:** With regard to the inclusion in the dataset of an earthquake of magnitude 3.2, which the EM-DAT catalogue reports to have caused significant damages, and which I suggested to remove from the regression dataset, the authors replied that, even though "M3-M3.9 earthquakes rarely causes any damage ... being rare is not being never" and that "Rarely causing damage means that there is at least a possibility of damage". However, since the main target of the proposed methodology is to compare hazard strength of different hazard types from their impact

for average exposure and vulnerability, I believe that the introduction of cases of abnormally high effect of small events could introduce a bias in the regression results.

**Response:** Thank you very much for your comment. We have added a new paragraph at the end of the Discussion section to address both this issue and the one raised in your next comment. We admit that inclusion or exclusion of data points such as this M3.2 earthquake may affect the modelling result, which deserve more attention in future work.

The new last paragraph of the Discussion section reads: "Beside these abovementioned issues, the inclusion and exclusion of certain data points based on values of variables may also affect the results of derivation of equivalency of hazard strength. First, in this study, a set of thresholds were adopted to filter out records of events with extremely small and large measures of magnitude indicators. However, some events with magnitude indicator measures barely inside the thresholds, such as the magnitude 3 earthquake in Southern Russia in 1999, were still included in the data for modelling. On the other hand, because the EM-DAT database only included events with loss records beyond a set of criteria, numerous events with lesser impacts were not recorded for model calibration in the study. Such exclusion of events with lesser impacts caused the empirical marginal distributions of the logarithmically transformed and standardized impact variables and the impact metric appear to be approximately Gaussian. Future work should explore to what extent the computation of equivalent hazard magnitude is sensitive to the inclusion and exclusion of data points of events of an either small or large size in terms of both the magnitude indicators and adverse impacts." (L400-409)

**Comment:** At this regard I noticed that, commenting Fig.2, the authors reported that "the empirical marginal distributions of the logarithmically transformed and standardized impact variables and the impact metric are approximately Gaussian". I find this observation rather suspect: there is no plausible reason why the occurrence of impact values should show a Gaussian distribution. On the contrary, one could expect that lower energy events causing lower effects should be more frequent, so that the distribution of the occurrence of impact values should show a decreasing trend as event energy and impact increase. This is well known in the case of earthquakes through the Gutenberg-Richter law, so that the deflection of the number of recorded events from a log-linear descending trend is used to identify the completeness threshold of datasets. However, similar relations were proposed for other types of hazardous events as well. Thus, the decrease of occurrences below the modal value is likely an artifact due to data collection incompleteness and to the adoption of a minimum cut-off threshold for impact variables (number of fatalities, number of people affected, etc.). Such incompleteness could introduce a bias in the regression results, since it is likely that, in the datasets, small events that caused effects larger than the average for similarly energetic events be over-represented. This, for instance, could be the case of the mentioned earthquake of magnitude 3. This problem should be at least discussed and possible checks of the effects of such bias could be attempted or at least proposed (e.g. comparing the regression results obtained with the entire dataset with those obtained for a subset where data below the modal value are removed).

**Response:** Thank you very much for your observation, comment, and suggestion. Indeed, it would make sense to assume that there is a lack of data points with small impact, which would make the distributions of impact variables look lognormal such that the transformed and standardized version of these variables would appear to be approximately Gaussian. To clarify this Gaussian

appearance of the empirical data, we have modified the first sentence of Section 4.1. To further address this issue, we have added sentences to the new last paragraph of the Discussion section as well.

The modified first sentence of Section 4.1 now reads: "Visualization of the distribution of data points with respect to the impact variables and impact metric (Figs. 2a, 2d, 2h, and 2m) shows that the empirical marginal distributions of the logarithmically transformed and standardized impact variables and the impact metric appear to be approximately Gaussian." (L185-187)

The new last paragraph of the Discussion section reads: "Beside these abovementioned issues, the inclusion and exclusion of certain data points based on values of variables may also affect the results of derivation of equivalency of hazard magnitude. First, in this study, a set of thresholds were adopted to filter out records of events with extremely small and large measures of magnitude indicators. However, some events with magnitude indicator measures barely inside the thresholds, such as the magnitude 3 earthquake in Southern Russia in 1999, were still included in the data for modelling. On the other hand, because the EM-DAT database only included events with loss records beyond a set of criteria, numerous events with lesser impacts were not recorded for model calibration in the study. Such exclusion of events with lesser impacts caused the empirical marginal distributions of the logarithmically transformed and standardized impact variables and the impact metric appear to be approximately Gaussian. Future work should explore to what extent the computation of equivalent hazard magnitude is sensitive to the inclusion and exclusion of data points of events of an either small or large size in terms of both the magnitude indicators and adverse impacts." (L400-409)

**Comment:** Once clarified these points, my opinion is that the manuscript could be accepted for *publication*.

**Response:** Thank you very much for your encouraging comment. We look forward to the editor's positive decision.

**Anonymous Referee #4**

We thank you very much for your constructive comments, questions, and suggestions. In the following, we copy your comments in *italics* and follow with our response. The major changes to the manuscript are summarized as follows:

- 1) We have modified the Discussion section to address the referees' comments.
- 2) We have corrected the typos.
- 3) We have updated the references.

**Comment:** Authors propose in this manuscript an equivalent hazard magnitude scale to measure the strength of natural force involved in the entirety of a natural hazard event for comparative analysis across different hazard types. The study is interesting but there are some issues which need to be clarified.

**Response:** Thank you very much for your summary and encouragement. We have made minor revisions to the manuscript, and we address your concerns as follows.

**Comment:** *Lines* 103-105: 12 hazard types have been taken into account, and namely: cold wave, convective storm, drought, earthquake, extra-tropical storm, flash flood, forest fire, heat wave, riverine flood, tornado, tropical cyclone, 105 and tsunami. How were the types of hazards selected? What are the criteria at the base of this choice? For what reasons other types (for instance, landslides, sinkholes, etc.) have been excluded? This part needs to be better clarified.

**Response:** Thank you very much for your comment. In the first paragraph of the Methodology section, we briefly introduced how the section was laid out. Regarding the selection of 12 hazard types, we provided the details of the rationale in the first paragraph of Section 3.1 Data Collection. Originally, we downloaded all available data from the EM-DAT database. However, many of the hazard types did not have magnitude indicator measures. Therefore, we could only include 12 hazard types with sufficient amount of data points with magnitude indicator values for modelling.

To explain this issue, the corresponding sentences in Section 3.1 read: "For this study, we downloaded the entire EM-DAT datasets on all types of natural hazards. However, since some records of hazard magnitude indicators of events for some hazard types (e.g., the volcanic activities and landslides) were missing, we only included 12 hazard types." (L116-119)

**Comment:** Lines 367-369: Authors state that "To demonstrate the implementation of the proposed methodology for deriving equivalent hazard magnitudes of events, we only considered one hazard magnitude indicator for each hazard type. For many hazard types, one indicator cannot represent the true hazard magnitude of an event which may arise due to multiple forcings". This latter sentence is very true, and is a strong assumption, which would deserve some more comments from the Authors: how much does this assumption influence the outcomes? Did they make attempts in evaluating the effects on the outcomes of such a choice?

**Response:** Thank you very much for your comment. We are glad that the referee also thinks that the consideration of multiple magnitude indicators is an important issue. If sufficient data on

multiple hazard magnitude indicators becomes available for modelling equivalent hazard magnitude on the Gardoni Scale, we would definitely be excited to conduct research to answer the questions raised by the referee. At the current stage, however, we only have one magnitude indicator for each hazard type because of the adoption of the datasets from the EM-DAT database. That is also mainly why we emphasized that this lack of data on other magnitude indicators was a limitation in the Discussion section.

**Comment:** *I* also agree with the comment from the other reviewer about the Gaussian distribution invoked by Authors when commenting figure 2. This, too, needs further explanation.**

**Response:** Thank you very much for your comment and suggestion. As mentioned in our response to the other referee, we agree that the empirical distribution of the logarithmically transformed and standardized impact variables and impact metric appeared to be Gaussian. Correspondingly, the original impact variables would appear to have a lognormal distribution due to lack of data on small impact measures. To highlight this Gaussian appearance of the empirical data, not the actual theoretical distribution of the impact variables and impact metric, we have modified the first sentence of Section 4.1. To further address this issue, we have added sentences to the new last paragraph of the Discussion section as well.

The modified first sentence of Section 4.1 now reads: "Visualization of the distribution of data points with respect to the impact variables and impact metric (Figs. 2a, 2d, 2h, and 2m) shows that the empirical marginal distributions of the logarithmically transformed and standardized impact variables and the impact metric appear to be approximately Gaussian." (L185-187)

The new last paragraph of the Discussion section reads: "Beside these abovementioned issues, the inclusion and exclusion of certain data points based on values of variables may also affect the results of derivation of equivalency of hazard magnitude. First, in this study, a set of thresholds were adopted to filter out records of events with extremely small and large measures of magnitude indicators. However, some events with magnitude indicator measures barely inside the thresholds, such as the magnitude 3 earthquake in Southern Russia in 1999, were still included in the data for modelling. On the other hand, because the EM-DAT database only included events with loss records beyond a set of criteria, numerous events with lesser impacts were not recorded for model calibration in the study. Such exclusion of events with lesser impacts caused the empirical marginal distributions of the logarithmically transformed and standardized impact variables and the impact metric appear to be approximately Gaussian. Future work should explore to what extent the computation of events of an either small or large size in terms of both the magnitude indicators and adverse impacts." (L400-409)

**Comment:** For all the above, I ask for minor revisions.**

**Response:** Thank you very much for your comment and suggestion. We look forward to the editor's decision.